# U-Turn Diffusion

## Abstract

We present a comprehensive examination of score-based diffusion models of AI for generating synthetic images. These models hinge upon a dynamic auxiliary time mechanism driven by stochastic differential equations, wherein the score function is acquired from input images. Our investigation unveils a criterion for evaluating efficiency of the score-based diffusion models: the power of the generative process depends on the ability to de-construct fast correlations during the reverse/de-noising phase. To improve the quality of the produced synthetic images, we introduce an approach coined "U-Turn Diffusion". The U-Turn Diffusion technique starts with the standard forward diffusion process, albeit with a reduced duration compared to conventional settings. Subsequently, we execute the standard reverse dynamics, initialized with the concluding configuration from the forward process. This U-Turn Diffusion procedure, combining forward, U-turn, and reverse processes, creates a synthetic image approximating an independent and identically distributed (i.i.d.) sample from the probability distribution implicitly described via input samples. To analyze relevant time scales we employ various analytical tools, including auto-correlation analysis, weighted norm of the score-function analysis, and Kolmogorov-Smirnov Gaussianity test. The tools guide us to establishing that analysis of the Kernel Inception Distance, a metric comparing the quality of synthetic samples with real data samples, reveals the optimal U-turn time.

## 1 Introduction

The fundamental mechanics of Artificial Intelligence (AI) encompass a three-step process: acquiring data, modeling it, and then predicting or inferring based on the constructed model. Culmination of this process is the generation of synthetic data, which serves as a core component of the prediction step.

Synthetic data holds the potential to augment information in various ways. It achieves this by leveraging model-derived conjectures to enrich the data's complexity and structure. In particular, Score Based Diffusion (SBD) models Song et al. (2021); Rombach et al. (2022); Ho et al. (2020) have emerged as a highly successful paradigm in this context. The foundation of the SBD models success rests on the notion that their inherent structure extracts a substantial amount of information from the data.

The essence of SBD models is deeply rooted in the concepts that the reality represented via data can emerge from noise or chaos, suggesting a process akin to de-noising (reverse part of the SBD dynamics), and that the introduction of diffusion can disrupt existing order within data (direct part of the SBD dynamics). These fundamental principles underlie the audacious approach of building generative models upon these very principles.

While the achievements of SBD models are impressive, they are not universally successful. Instances where barriers are significant, referred to colloquially in physics jargon as "glassy" scenarios Charbonneau et al. (2023), may necessitate the graceful addition of diffusion and bias/advection possibly nonlinear (in the state space), compelling the extension of SBD model runtime for better performance.

Our overarching objective revolves around gaining insights into existing successful SBD models and further enhancing their capabilities. We methodically approach this goal by breaking it down into steps. However, in this manuscript our primary focus resides not in refining the diffusion component of the model. Instead, we presume this component as given to us, as already developed and documented in prior works (e.g., Song et al.

(2021); Rombach et al. (2022); Ho et al. (2020)). Then our attention centers on comprehending temporal correlations within both the diffusion process (forward part of SBD) and the denoising/reconstruction process (reverse part of SBD).

A pricipal outcome of our analysis of temporal correlations is a fundamental realization concerning the optimal termination point of the forward process, i.e. of the U-Turn point. This culminates in the proposal of a novel algorithm termed "U-Turn Diffusion." This algorithm provides guidance on when to pivot from the direct to the reverse process. Moreover, we naturally initialize the reverse process after the U-turn with the last configuration of the forward process.

In summary, this manuscript presents a comprehensive exploration of the dynamics of SBD models, delving into details of the temporal correlations that underpin their success. Our insights not only enhance the understanding of these models but also lay the foundation for the development of novel techniques, such as the U-Turn Diffusion algorithm, which promises to further elevate the capabilities of SBD-based generative modeling.

The manuscript is structured as follows: In Section 2, we provide a technical introduction, laying the foundation by outlining the construction of SBD models. Section 3 forms the first original contribution of this work, encompassing an extensive correlation analysis. We delve into two-time auto-correlation functions of the SBD, establishing relevant time scales. Additionally, we identify the emergence of similar time scales in single-time tests of (a) the average 2-norm of the score-function and (b) the Kolmogorov-Smirnov criterion for Gaussianity. This section reaches its climax with the proposal of the U-Turn diffusion algorithm, discussed in Section 4. Our manuscript concludes by summarizing findings and outlining future directions for research in Section 6.

## 2 Technical Introduction: Setting the Stage

Within this manuscript, we embrace the Score-Based Diffusion (SBD) framework, as expounded in Song et al. (2021). The SBD harmoniously integrates the principles underlying the "Denoising Diffusion Probabilistic Modeling" framework introduced in Sohl-Dickstein et al. (2015) and subsequently refined in Ho et al. (2020), along with the "Score Matching with Langevin Dynamics" approach introduced by Song & Ermon (2019). This seamless integration facilitates the reformulation of the problem using the language of stochastic differential equations, paving the way to harness the Anderson's Theorem Anderson (1982). As elucidated in the following, this theorem assumes a principal role in constructing a conduit linking the forward and reverse diffusion processes.

Let us follow Song et al. (2021) and introduce the forward-in-time Stochastic Ordinary Differential Equation (SDE):

$$d\mathbf{x}_t = f(\mathbf{x}_t, t)dt + G(\mathbf{x}_t, t)d\mathbf{w}_t, \tag{1}$$

and another reverse-in-time SDE:

$$d\mathbf{x}_t = \left(f(\mathbf{x}_t, t) - \nabla.(G(\mathbf{x}_t, t)G(\mathbf{x}_t, t)^T) - G(\mathbf{x}_t, t)G(\mathbf{x}_t, t)^T \left(\nabla_{\mathbf{x}_t} \log\left(p_t(\mathbf{x}_t)\right)\right)\right) dt + G(\mathbf{x}_t, t)d\bar{\mathbf{w}}_t, \tag{2}$$

where the drift/advection $f : \mathbb{R}^{n_x} \times \mathbb{R} \to \mathbb{R}^{n_x}$ and diffusion $G : \mathbb{R}^{n_x} \times \mathbb{R} \to \mathbb{R}^{n_x} \times \mathbb{R}^{n_x}$ are sufficiently smooth (Lipschitz functions). Additionally, we assume the existence of a well-defined initial distribution $p_0(\cdot)$ represented by data (samples), and both forward and backward processes are subject to Ito-regularization. The Wiener processes $\mathbf{w}_t$ and $\bar{\mathbf{w}}_t$ represent standard Wiener processes for forward and reverse in time, respectively.

Anderson's theorem establishes that the forward-in-time process and the reverse-in-time process have the same marginal probability distribution, denoted by $p_t(\cdot)$.

*Remark.* The proof of Anderson's Theorem relies on the equivalence of the Fokker-Planck equations derived for the direct (1) and inverse (2) dynamics:

$$\partial_t p_t(\mathbf{x}) - \nabla_{\mathbf{x}}\left(f(\mathbf{x},t)p_t(\mathbf{x})\right) = \frac{1}{2}\nabla_{\mathbf{x}}\left(G(\mathbf{x},t)G(\mathbf{x},t)^T\nabla_{\mathbf{x}}p_t(\mathbf{x})\right),$$

$$\partial_t p_t(\mathbf{x}) - \nabla_{\mathbf{x}}\left(f(\mathbf{x},t)p_t(\mathbf{x})\right) + \nabla_{\mathbf{x}}\left(G(\mathbf{x},t)G(\mathbf{x},t)^T\left(\nabla_{\mathbf{x}}\log\left(p_t(\mathbf{x})\right)\right)\right)p_t(\mathbf{x}) = -\frac{1}{2}\nabla_{\mathbf{x}}\left(G(\mathbf{x},t)G(\mathbf{x},t)^T\nabla_{\mathbf{x}}p_t(\mathbf{x})\right),$$

where $p_t(\cdot)$ is the marginal probability distribution of $\mathbf{x}$ which is equivalent for the forward and inverse processes (by construction).

The forward diffusion process transforms the *initial distribution* $p_0(\cdot)$, represented by samples, into a *final distribution* $p_T(\cdot)$ at time $T$. The terms $f(\mathbf{x},t)$ and $G(\mathbf{x},t)$ in the SDE are free to choose, but in the SBD approach, they are selected in a data-independent manner such that $p_T(\cdot)$ converges to $\pi(\cdot) = \mathcal{N}(\cdot,\mathbf{0},\mathbf{I})$ as $T$ approaches infinity. This convergence ensures that the generated samples align with a target distribution, typically the standard normal distribution $\mathcal{N}(\cdot,\mathbf{0},\mathbf{I})$.

**Inference**, which involves generating new samples from the distribution represented by the data, entails initializing the reverse process (2) at $t = T$ (large but finite) with a sample drawn from $\pi(\cdot)$, and then running the process backward in time to reach the desired result at $t = 0$. This operation requires accessing the so-called *score function* $s(\mathbf{x},t) = (\nabla_{\mathbf{x}}\log(p_t(\mathbf{x}))$, as indicated in Eq. (2). However, practically obtaining the exact time-dependent score function is challenging. Therefore, we resort to approximating it with a Neural Network (NN) parameterized by a vector of parameters $\theta$: $s_\theta(\mathbf{x},t) \approx s(\mathbf{x},t)$.

The neural network-based approximation of the score function allows us to efficiently compute and utilize gradients with respect to the input data $\mathbf{x}$ at different times $t$, which is essential for guiding the reverse process during inference. By leveraging this neural network approximation, we can effectively sample from the desired distribution and generate new images which are approximately i.i.d. from a target probability distribution represented by input data. This approach enables us to achieve reliable and accurate inference in complex high-dimensional spaces, where traditional methods may struggle to capture the underlying data distribution effectively.

**Training:** The neural network $s_\theta(\mathbf{x}_t,t)$ can be trained to approximate the score function $\nabla_{\mathbf{x}_t}\log p_t(\mathbf{x}_t)$ using the weighted De-noising Score Matching (DSM) objective Song et al. (2021):

$$\mathcal{L}_{DSM}(\theta,\lambda(\cdot)) := \frac{1}{2}\mathbb{E}_{\substack{t\sim U(0,T),\\ \mathbf{x}_0\sim p_0(\mathbf{x}_0),\\ \mathbf{x}_t\sim\sim p_t(\mathbf{x}_t|\mathbf{x}_0)}}[\lambda(t)\|\nabla_{\mathbf{x}_t}\log p_t(\mathbf{x}_t|\mathbf{x}_0) - s_\theta(\mathbf{x}_t,t)\|_2^2], \qquad (3)$$

This approach offers significant advantages over alternative methods, such as those described in Hyvärinen (2005); Vincent (2011), due to the analytical evaluation of $p_t(\mathbf{x}_t|\mathbf{x}_0)$ as an explicit function of $\mathbf{x}_0$ for various simple drift and diffusion choices in the forward SDE. The objective function $\mathcal{L}_{DSM}$ leverages the score matching technique to ensure that the gradients estimated by the neural network closely align with the true gradients of the log-likelihood. The weight function $\lambda(t)$ allows us to assign varying importance to different time points during training, offering further flexibility in optimizing the neural network's performance.

In the two subsequent subsections, we will explore the freedom in selecting the drift/advection and diffusion terms in the forward SDE, as well as the implications of choosing specific weight functions $\lambda(t)$. This analysis will provide valuable insights into the overall performance of the Score-Based Diffusion (SBD) framework and its impact on improving the scheme.

## 2.1 Variance Preserving SDE

In this manuscript we focus on time inhomogeneous Ornstein Uhlenbeck Stochastic Differential Equations (SDEs) known as Variance Preserving (VP) Stochastic Differential Equations (VP-SDEs) that possess closed-form solutions. The capability to uncover a closed-form solution is essential for learning efficiency, as it allows for the evaluation of each sample just once, encompassing automatically all potential paths within the SDE that adhere to the initial condition set by the sample. We achieve this by choosing specific drift and diffusion

functions as follows:

$$f(\mathbf{x}_t, t) = -\frac{1}{2}\beta(t)\mathbf{x}_t, \quad G(\mathbf{x}_t, t) = \sqrt{\beta(t)}. \tag{4}$$

This choice results in the following form of the VP-SDE (1) supplemented by the initial conditions:

$$d\mathbf{x}_t = -\frac{1}{2}\beta(t)\mathbf{x}_t dt + \sqrt{\beta(t)}\ d\mathbf{w}_t, \quad \mathbf{x}_0 \sim p_{\text{data}}. \tag{5}$$

Here, $\beta(t)$ is a positive function, often referred to as the noise scheduler. We chose this specific form for the drift and diffusion functions based on considerations of simplicity and historical context. Linearity in Eq. (5) was a practical consideration that allows us to express $\mathbf{x}_t$ analytically in terms of $\mathbf{x}_0$. The affine drift in $\mathbf{x}$ and $\mathbf{x}$-independent diffusion, as opposed to more general linear forms in $\mathbf{x}$ for both, were inherited from their original discrete-time counterparts.

The original discrete version of the VP-SDE was given by:

$$\mathbf{x}_n = \sqrt{1 - b_n}\mathbf{x}_{n-1} + \sqrt{b_n}\mathbf{z}_{n-1}, \ \mathbf{z}_n \sim \mathcal{N}(\mathbf{0}, \mathbf{I}), \tag{6}$$

where $n = 1, \cdots, N$. By introducing $\beta \doteq b/\Delta$, $t \doteq \Delta n$, $T \doteq \Delta N$, and taking the limit: $N \to \infty$, $\Delta \to 0$, Eq.(6) transforms into Eq.(5).

The solution to Eq. (5) is given by:

$$\mathbf{x}_t = \sqrt{\Phi(t,0)}\ \mathbf{x}_0 + \int_0^t \sqrt{\Phi(t,s)\beta(s)}\ d\mathbf{w}_s, \tag{7}$$

where $\Phi(t,s) = e^{-\int_s^t \beta(u)\mathrm{d}u}$ and $\mathbf{x}_t$ conditioned on $\mathbf{x}_0$ is Gaussian:

$$p(\mathbf{x}_t|\mathbf{x}_0) = \mathcal{N}\left(\mathbf{x}_t; \sqrt{\Phi(t,0)}\mathbf{x}_0, (1 - \Phi(t,0))\hat{\mathbf{I}}\right). \tag{8}$$

For completeness, we present a set of useful formulas derived from Eq. (8) that describe correlations within the forward process:

$$\mathbb{E}[\mathbf{x}_t] = \sqrt{\Phi(t,0)}\ \mathbb{E}[\mathbf{x}_0], \tag{9}$$

$$\mathbb{E}[\mathbf{x}_t^2] = \Phi(t,0)\ \mathbb{E}[\mathbf{x}_0^2] + \int_0^t \Phi(t,s)\beta(s)ds = \Phi(t,0)\left(\mathbb{E}[\mathbf{x}_0^2] - 1\right) + 1, \tag{10}$$

$$\mathbb{V}[\mathbf{x}_t] = \mathbb{E}[\mathbf{x}_t^2] - \mathbb{E}[\mathbf{x}_t]^2 = 1 - \Phi(t,0), \tag{11}$$

$$\mathbb{E}[\mathbf{x}_t\mathbf{x}_s] = \sqrt{\Phi(\xi,\gamma)}\ \mathbb{E}[\mathbf{x}_\gamma^2], \quad \xi = \max(t,s), \ \gamma = \min(t,s) \tag{12}$$

$$C_0(t) \doteq \frac{\mathbb{E}[\mathbf{x}_0\mathbf{x}_t]}{\mathbb{E}[\mathbf{x}_0^2]} = \sqrt{\Phi(t,0)}, \tag{13}$$

$$C_T(t) \doteq \frac{\mathbb{E}[\mathbf{x}_T\mathbf{x}_t]}{\mathbb{E}[\mathbf{x}_T^2]} = \sqrt{\Phi(T,t)}\ \frac{\mathbb{E}[\mathbf{x}_t^2]}{\mathbb{E}[\mathbf{x}_T^2]}, \tag{14}$$

where $\mathbb{E}[\cdots]$ and $\mathbb{V}[\cdots]$ represent expectations and variances over the forward VP process (5). By shifting and re-scaling the initial data to ensure that $\mathbb{E}[\mathbf{x}_0] = 0$ and $\mathbb{E}[\mathbf{x}_0^2] = 1$, we find from Eqs. (9, 10, 11) that $\mathbb{E}[\mathbf{x}_t] = 0$, and $\mathbf{x}_t$ indeed becomes Variance Preserving (VP), as the name suggests, since $\mathbb{E}[\mathbf{x}_t^2] = 1$.

It is important to note that while the direct process (5) only depends on $\mathbf{x}_0$ through the initial condition, resulting in the explicit solution (7), the respective reverse process described by Eq.(2), with fixed $f(\cdot)$ and $G(\cdot)$ according to Eq.(4), carries information about $\mathbf{x}_0$ through the score function.

## 2.2 Re-weighting in the Score Function Training

In the context of Eq. (3), choosing an appropriate weight function $\lambda(t)$ is crucial. While there are various options for $\lambda(t)$ (as discussed in Karras et al. (2022); Kingma & Gao (2023) ), we adopt in this work the approach introduced in Song & Ermon (2019). Specifically, we substitute:

$$\lambda(t) \to 1 - \Phi(t, 0). \tag{15}$$

This choice of $\lambda(t)$ is well-motivated as it accounts for the scaling with the $\mathbf{x}_0$-independent part of the conditional probability described by Eq. (8). To elaborate, we estimate the term involving $\lambda(t)$ in the objective function as follows:

$$\lambda(t)|\nabla_{\mathbf{x}_t}\log p_t(\mathbf{x}_t|\mathbf{x}_0) - s_\theta(\mathbf{x}_t; t)|^2 \sim \lambda(t)|\nabla_{\mathbf{x}_t}\log p_t(\mathbf{x}_t|\mathbf{x}_0)|^2 \sim \lambda(t)\frac{|\mathbf{x}_t - \sqrt{\Phi(t, 0)}\mathbf{x}_0|^2}{(1 - \Phi(t, 0))^2} \sim \frac{\lambda(t)}{1 - \Phi(t, 0)},$$

This suggests that by choosing $\lambda(t)$ according to Eq. (15), we equalize the contributions from different time steps into the integration (expectation) over time in the objective function (3). This ensures a balanced influence from all time steps, making the learning process more effective and efficient.

By incorporating the selected $\lambda(t)$, our approach successfully captures the inherent characteristics of the VP-SDE solution and leverages them for generative modeling.

# 3 Numerical Experiments in the Standard Setting

In this section, we present the results of our numerical experiments, which involve different direct and reverse processes defined in Eq.(5) (or Eq.(7)) and Eq. (2), with the specific choices of Eq. (4). Additionally, we explore various profiles for the function $b(n)$, which are introduced in the following. These profiles allow us to test the sensitivity and effectiveness of the methods under varying balance between advection and diffusion. By systematically exploring these setups, we gain valuable insights into the generative capabilities and limitations of the models based on the VP-SDE formulation.

The experimental findings presented below shed light on the interplay between the direct and reverse processes, revealing how they collectively contribute to the overall generative performance, then suggesting how to improve the process.

## 3.1 Profiles of $b$ (discrete version of $\beta$)

| Profile of $b$ | Definition |
|---|---|
| Linear Ho et al. (2020) | $b(n) = (b_2 - b_1)\frac{n}{N} + b_1, n \in [0, N]$ |
| Cosine Nichol & Dhariwal (2021) | $b(n) = \min\left(1 - \frac{p(n)}{p(0) - p(n)}, 0.999\right),\ p(n) = \cos^2\left(\frac{n/N + 0.008}{1 + 0.008} \cdot \frac{\pi}{2}\right)$ |
| Sigmoid Xu et al. (2022) | $b(n) = \frac{b_2 - b_1}{1 + \exp(-12n/N + 6)} + b_1, n \in [0, N]$ |

Table 1: Profiles of $b$ (discrete version of $\beta$ associated with the discretization of time $t$ using integer $n$, where $t = n\Delta$) commonly employed in previous studies Ho et al. (2020); Nichol & Dhariwal (2021); Xu et al. (2022), and also adopted for our investigation. For our discrete-time simulations (as discussed in the following), we set $b_1 = 0.0001$ and $b_2 = 0.02$, and consider integer values of $n$ in the range $0, 1, \cdots, N = 1000$. These profiles have been selected to enable a comprehensive assessment of our approach and are representative of the noise conditions commonly used in the literature.

Fig.(1) displays the mean and standard deviation (square root of variance) of the direct process samples $x_n$, which are $\sqrt{\Phi(n\Delta, 0)}$ and $\sqrt{1 - \Phi(n\Delta, 0)}$ respectively, as described by Eq.(8). These results are presented for three distinct Variance Dampening (VD) profiles of $b(n)$ – linear, sigmoid, and cosine – as outlined in Table 1.

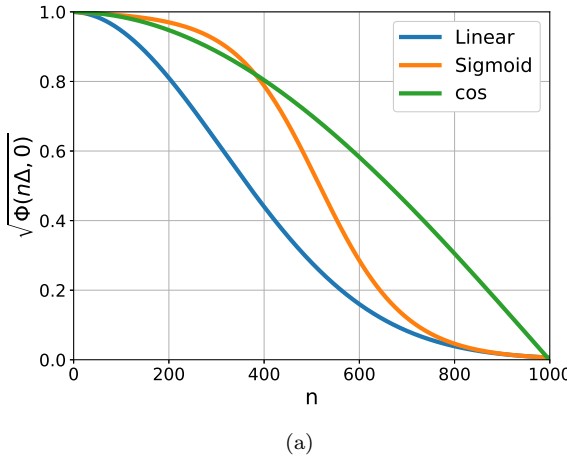 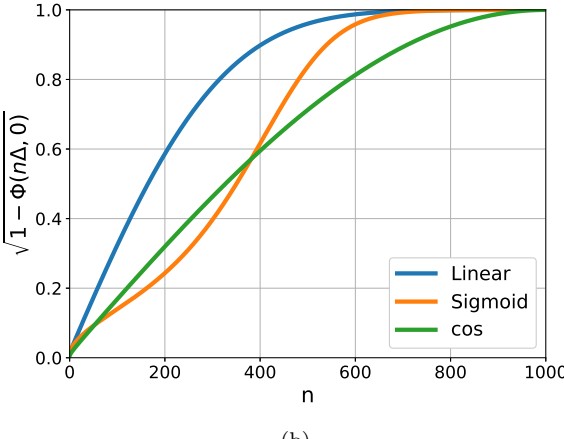

(a)                                                           (b)

Figure 1: Mean and standard deviation of the direct process samples $x_t$ for three different Variance Damp-ening (VD) protocols of $b(n)$ – linear, sigmoid, and cosine. The $x$-axis represents time $t$, and the $y$-axis corresponds to the mean and standard deviation of the direct process samples $x_t$.

The original design of $b(n)$ was motivated by the desire to achieve a smooth transition forward in time from the part of the dynamics, $x_n$, that retains information about the initial condition $x_0$ (dominated by the drift or ballistic dynamics) to a phase where this information is gradually forgotten (dominated by diffusion). This trend is clearly evident in Figs.(1a,b), in line with the behavior described by Eqs.(9) and (11). Among the three VD profiles, the linear one exhibits the most rapid decrease/increase in drift/diffusion over time, while the cos profile results in a more gradual and slower transition.

### 3.2   Samples of the forward and backward processes

Here we explore both the forward process $(\mathbf{x}(t)|t \in [0, T])$ and the reverse process $(\mathbf{x}_r(t)|t \in [0, T])$. For consistency and clarity, we use the same notation for time in both processes, following a forward counting scheme. In our simulations, we discretize time, taking values in the range $0, \cdots, T = 1000$ to facilitate numerical computations and analysis.

Figures (2) and (3) showcase temporal samples of both the forward and reverse processes, each corresponding to the three different profiles of $b$. Notably, we observe that the linear and cosine profiles exhibit the most rapid and gradual decay, respectively, among the three options.

### 3.3   Auto-Correlation Functions of the Reverse Process

In our analysis of the Score-Based-Diffusion (SBD) method, we conduct computations by averaging (comput-ing expectations) over multiple samples of the stochastic processes. Our focus is on studying auto-correlation functions, as they serve as direct indicators of how the processes retain or discard information over time.

The auto-correlation functions of the forward process are fully described in Eqs.(13) and (14). Therefore, nu-merical experiments for the forward process serve primarily as a sanity check, since the analytical expressions are available. However, for the reverse VP process, described by Eq.(2) with drift and diffusion functions according to Eq.(4), no analytical expressions are available for the auto-correlation functions. Consequently, we primarily investigate these auto-correlation functions numerically.

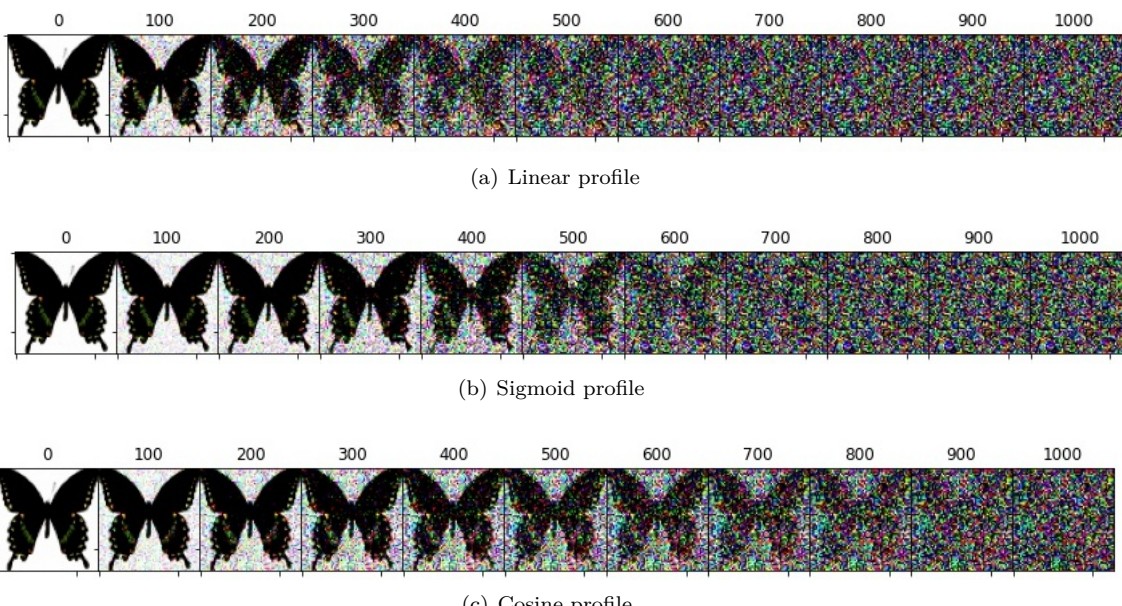

Figure 2: Temporal samples of the forward process $x_n$ shown for three different noise profiles described in Table 1. Each figure represents a distinct noise profile, demonstrating the dynamic behavior of the process over time.

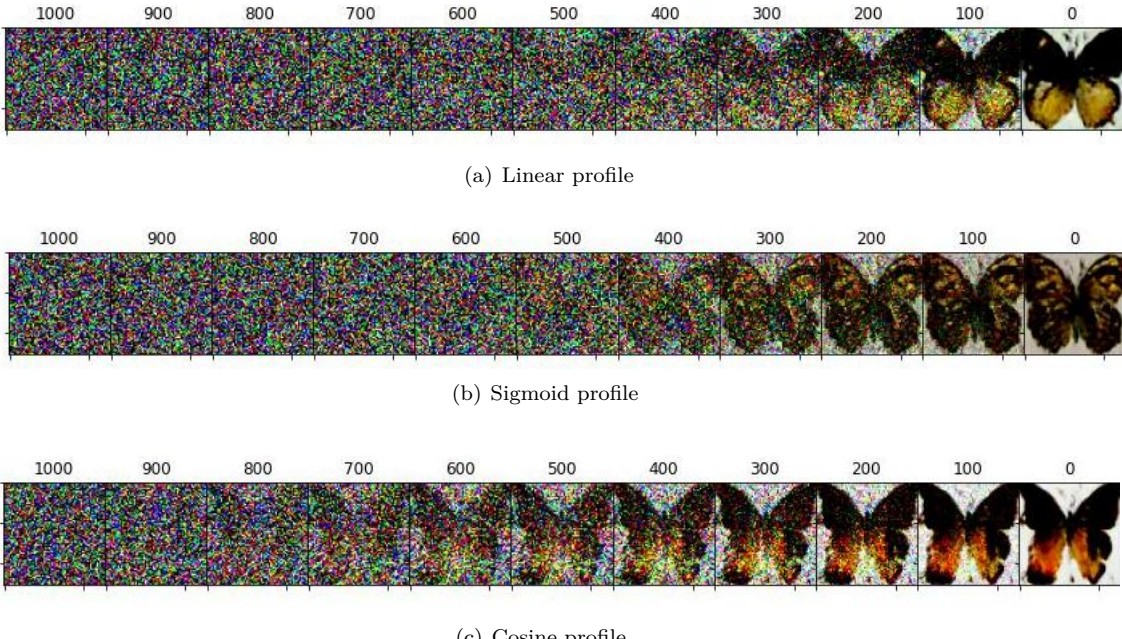

Figure 3: Temporal samples of the reverse process for three different noise profiles described in Table 1. Each figure corresponds to a distinct noise profile, providing a representation of the dynamic behavior of the reverse process over time.

Specifically, we study the auto-correlation functions of the reverse process between the current time $\tau$ and an earlier or later reference time $t$:

$$C_{t;r}(\tau) = \frac{\mathbb{E}[\mathbf{x}_r(t)\mathbf{x}_r(\tau)]}{\mathbb{E}[(\mathbf{x}_r(t))^2]}, \tag{16}$$

where $\tau$ can be smaller or larger than $t$. These auto-correlation functions provide valuable insights into the behavior of the reverse process and its ability to recall information from earlier time steps, contributing to a comprehensive understanding of the generative capabilities of the SBD method.

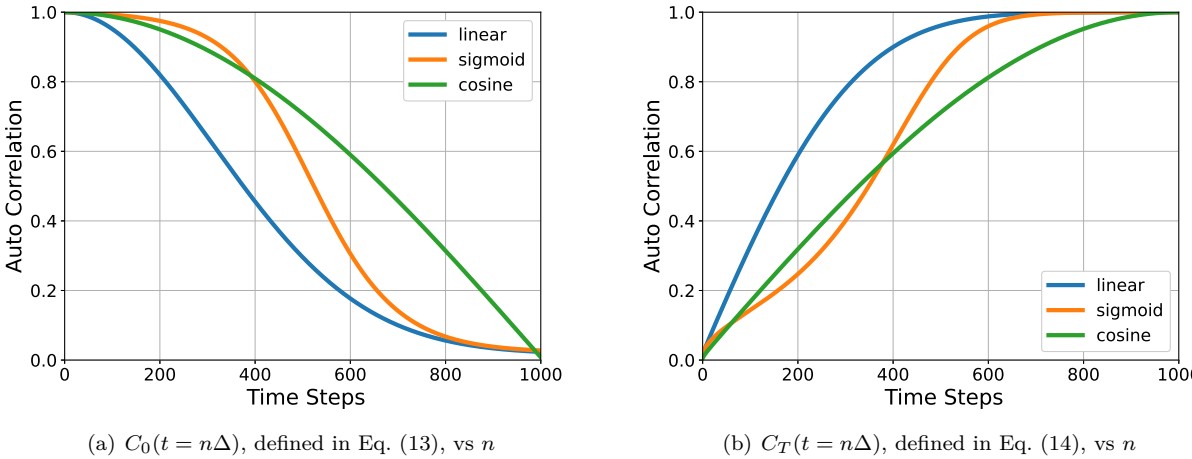

(a) $C_0(t = n\Delta)$, defined in Eq. (13), vs $n$      (b) $C_T(t = n\Delta)$, defined in Eq. (14), vs $n$

Figure 4: Auto correlation for forward process.

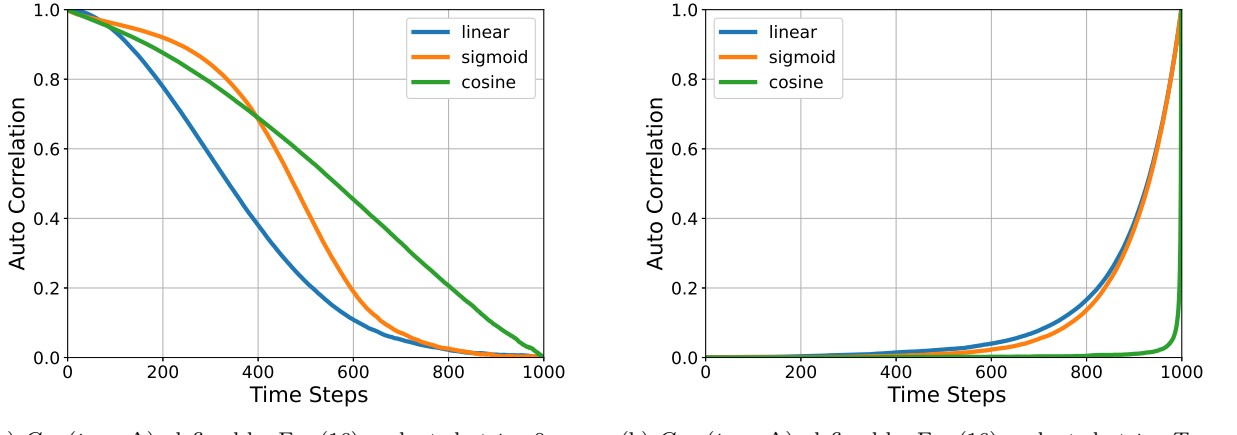

(a) $C_{0;r}(t = n\Delta)$, defined by Eq. (16) evaluated at $t = 0$, vs $n$    (b) $C_{T;r}(t = n\Delta)$, defined by Eq. (16) evaluated at $t = T$, vs $n$

Figure 5: Auto correlation for reverse process.

The auto-correlation function analysis results for three different advection/diffusion (noise) profiles are presented in Fig.(4) and Fig.(5) for the forward and reverse processes, respectively. These findings yield several important observations:

1. The auto-correlation functions demonstrate clear differences among the various processes, supporting the notion of using auto-correlation as an indicator of "correlation" decay, i.e., how quickly the signals lose correlation (information) over time. Among the three forward processes, the "linear" and "cosine" profiles exhibit the fastest and slowest decay of correlations, respectively, which is consistent with the temporal evolution of samples shown in Fig. (2).

2. Although correlations between $t = 0$ and subsequent times are destroyed/reconstructed similarly in both the forward and reverse processes, the correlations between $t = T$ and preceding moments of

time are remarkably different. Specifically, the $T$-referenced auto-correlation function of the reverse process, $C_{T;r}(t)$, decays much faster with decreasing $t$ compared to the auto-correlation function of the forward process, $C_T(t)$. This observation indicates that while the forward process retains all the original-sample-specific information in the initial conditions, the reverse process transforms this information into the "advection" term, spreading the information over time.

3. Moreover, the decay of correlations in the reverse process counted from $T$ is the fastest for the cosine profile. This finding implies that by engineering the forward process (independent of the initial conditions), we can achieve a faster or slower decay of correlations in the reverse process.

4. Furthermore, the dramatic decay of correlations in the reverse process, as observed in $C_{T;r}(t)$, indicates that the information contained in $\mathbf{x}(T)$ is rapidly forgotten. To quantify this behavior, we study the $1/2$-decay correlation time $\delta(t)$, defined by

$$C_{t;r}(t \mp \delta(t)) = 1/2, \tag{17}$$

and its dependence on $t$, which is illustrated in Fig. (6).

In summary, these observations shed light on the distinctive behaviors of the forward and reverse processes, providing valuable insights into their information retention capabilities and temporal characteristics. The correlation decay analysis offers a deeper understanding of the generative dynamics underlying the SBD method.

While analyzing the correlation function provides a direct means to assess information retention during the forward and reverse processes, it lacks temporal locality. This absence of temporal locality poses challenges when incorporating auto-correlation functions into the algorithm(s), in particular the U-turn diffusion algorithm introduced in discussed in Section 4 below. Consequently, it motivates us to explore two alternative, time-local features in the upcoming two subsections.

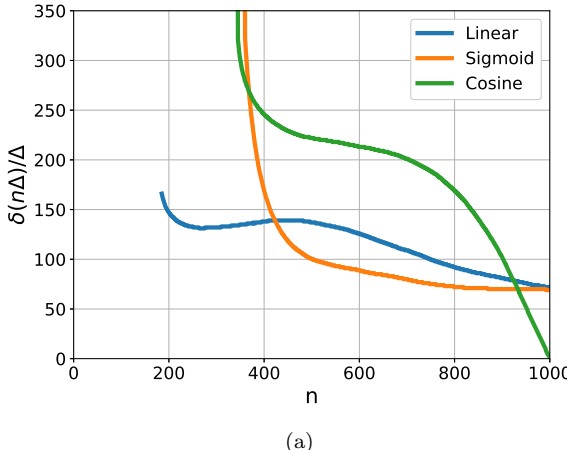
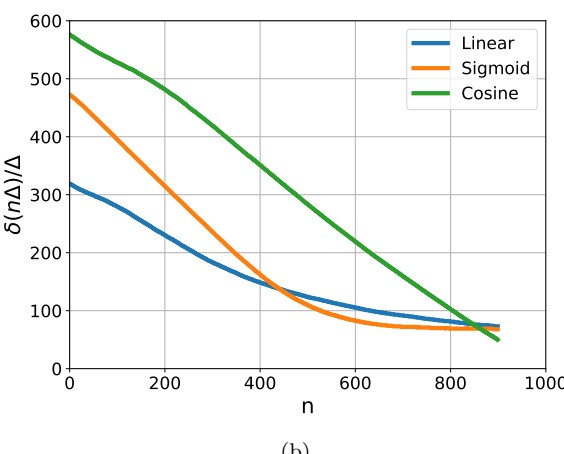

(a)          (b)

Figure 6: Dependence of the $1/2$-correlation time (discrete time version) $\delta(n\Delta)/\Delta$ of the reverse process, as defined in Eq. (17), on the (discrete) time index, $n$, for $\tau < t$, in (a), and for $\tau > t$, in (b).

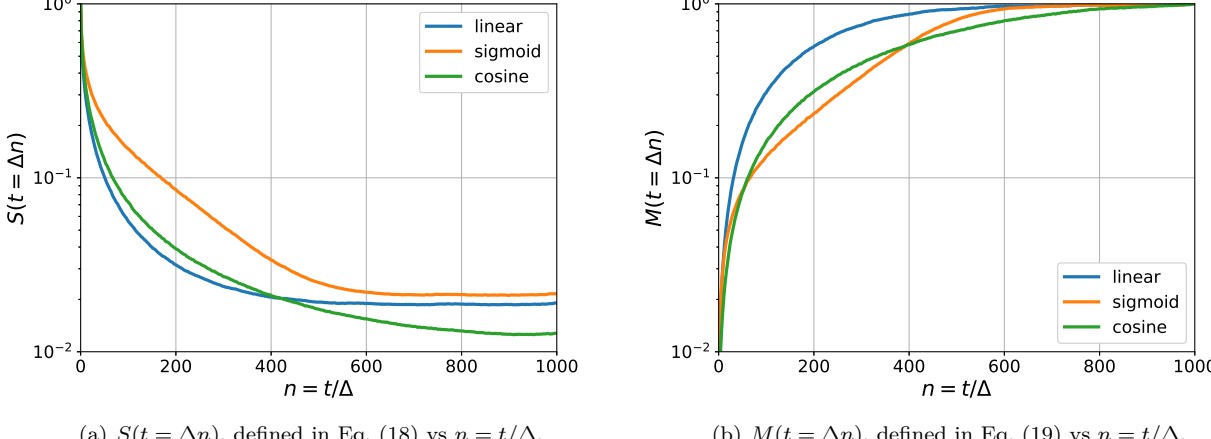

(a) $S(t = \Delta n)$, defined in Eq. (18) vs $n = t/\Delta$.

(b) $M(t = \Delta n)$, defined in Eq. (19) vs $n = t/\Delta$.

Figure 7: Average of the Score Function 2-Norm for the experiments described in the main text.

### 3.4 Average of the Score Function 2-Norm

The analysis of the time-dependence of the average of the score function (a vector) 2-norm is presented in Fig. (7a). This score function is denoted as:

$$S(t) \doteq \sqrt{\frac{\mathbb{E}\left[(\nabla_{\mathbf{x}} \log p_t(\mathbf{x}))^2\right]}{\mathbb{E}\left[(\nabla_{\mathbf{x}} \log p_0(\mathbf{x}))^2\right]}}, \tag{18}$$

In Fig. (7b), we present the average score function 2-norm, weighted with the $\sqrt{\lambda(t)}$-factor, and normalized at $t = T$ according to:

$$M(t) \doteq \sqrt{\frac{\lambda(t)\mathbb{E}\left[(\nabla_{\mathbf{x}} \log p_t(\mathbf{x}))^2\right]}{\lambda(T)\mathbb{E}\left[(\nabla_{\mathbf{x}} \log p_T(\mathbf{x}))^2\right]}}. \tag{19}$$

These figures illustrate how the score function norms, weighted and not, evolve over time, offering valuable insights into the generative modeling process and the relevance of the score function in capturing the underlying dynamics. The figures help to appreciate how the weighted score function provides a means to evaluate the importance of different time steps in the generative learning process. Specifically, we observe in Fig. 7, that the "mixed" regime establishes more rapidly: at $n \approx 500$ for the linear profile, $n \approx 600$ for sigmoid profile, and $n \approx 800$ for cosine profile. Notably, the score-function dynamics of the sigmoid profile also exhibit a marked shift at an earlier time of $n \approx 100$. We also note that the "correlation" times observed in the score-function profiles for different noise configurations approximately align with the "correlation" times depicted in Fig. 6b, where we analyzed the variation of the 1/2-correlation decay (for the reversed process, counted forward in time) in the auto-correlation function with respect to changes in the reference time.

### 3.5 Kolmogorov-Smirnov Gaussianity Test

We employ the Kolmogorov-Smirnov (KS) Gaussianity test to examine the null hypothesis: "Is a single-variable marginal of the resulting multi-variate distribution of $\mathbf{x}_r(t)$ at a given time $t$ Gaussian or not?" To validate this hypothesis, we apply the KS test to each of the single-variable marginals $p_t(x_k) = \int d(\mathbf{x} \setminus x_k) p_t(\mathbf{x})$. The KS-ratio is then calculated as the number of single dimensions $k$ for which the Gaussianity

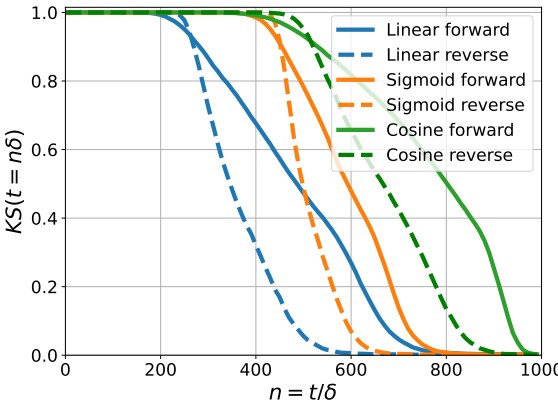

Figure 8: Kolmogorov-Smirnov test for forward and reverse dynamics under different $b$-protocols.

of the corresponding $x_k(t)$ is not confirmed, divided by the total number of dimensions (cardinality of $\mathbf{x}(t)$):

$$\text{KS}(t) = \frac{\text{\# of } p_t(x_k) \text{ failing the Gaussianity test}}{\text{cardinality of } \mathbf{x} = 64 \times 64 \times 3}.$$

The results of this analysis are displayed in Fig. 8.

Firstly, we note analyzing Fig. 8 that the "correlation" times from the KS profiles of the reverse process (across different noise schedules) show a close alignment with those presented in Fig. 6 for the shifts in the 1/2-correlation decay of the auto-correlation function in relation to variations in reference time.

Secondly, we notice that there's a noticeable misalignment in Fig. 8 between the forward and reverse KS curves. Theoretically, when considering appropriate limits, these curves should coincide. Several factors contribute to the score-function based estimation of the probability distribution constructed from the data, resulting in misalignment. Firstly, the Neural Network is only an approximate fit to the score function it approximates. Secondly, although the "reverse" Fokker-Planck equation is exact, ensuring the same marginal probability distribution as derived in the forward process, this exactness holds only in continuous time, while the actual implementation occurs in discrete time. Lastly, the number of input (data) samples, while assumed to be large, is still finite. All three errors accumulate, leading to the observed mismatch between the KS tests performed for the forward and reverse processes, as shown in Fig. (8). This discrepancy aligns with earlier observations and discussions reported in De Bortoli et al. (2021); Block et al. (2022). Additionally, we find that the mismatch in the KS-curves diminishes with improvements in the quality of the Neural Network, reduction of the number of discretization steps, and/or an increase in the number of input samples. These insights highlight the importance of these factors in refining the generative modeling process and reducing discrepancies between forward and reverse processes.

We opted for the KS test over the KL test for several reasons. Firstly, KL divergence, a measure of dissimilarity between distributions, is asymmetric and not truly a distance metric, whereas the KS test, which assesses the maximum separation between a test distribution and a reference distribution, is symmetric. Secondly, the KS test is non-parametric, meaning it doesn't depend on the parameters of the benchmark distribution (in our case, Gaussian), which would be a concern with KL divergence. Thirdly, KL divergence is more advantageous when comparing distributions with nontrivial tails, which isn't the focus of our analysis. Lastly, the KS test is computationally simpler.

### 3.6 Quality of Inference

Next we employ the Kernel Inception Distance (KID) Bińkowski et al. (2021) as our chosen metric to assess the quality of inference. KID measures the dissimilarity between the distributions of real and generated samples without assuming any specific parametric form for them. The KID is constructed by embedding

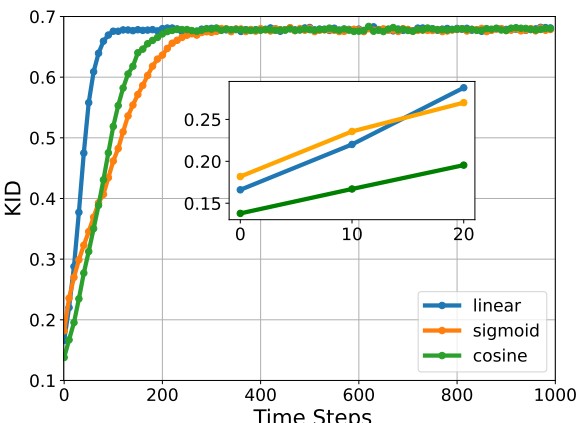

Figure 9: Kernel Inception Distance (KID) for the reverse process with different $b$-protocols displayed as a function of $n$. The inset provides a closer view of the primary trend at the lowest $n$ values, where the lowest KID values (best results) are attained. (We remind that the synthetic sample is generated at the end of the reverse process, i.e., at $n = 0$.)

real and generated images into a feature space using the Inception network Gretton et al. (2012). It then becomes a squared Maximum Mean Discrepancy (MMD) between the inception features of the real and generated images:

$$\text{KID}(\mathbb{P}_r, \mathbb{P}_g) = \mathbb{E}_{\substack{\mathbf{x}_r, \mathbf{x}'_r \sim \mathbb{P}_r, \\ \mathbf{x}_g, \mathbf{x}'_g \sim \mathbb{P}_g}} \left[ k(\mathbf{x}_r, \mathbf{x}'_r) + k(\mathbf{x}_g, \mathbf{x}'_g) - 2k(\mathbf{x}_r, \mathbf{x}_g) \right]$$

where $\mathbb{P}_r$ and $\mathbb{P}_g$ represent the distributions of real and generated samples, respectively and $k(\mathbf{x}, \mathbf{y}) = (\mathbf{x}^\top \mathbf{y} + 1)^3$ is a polynomial kernel. The KID quantifies the distance between the two distributions, with a lower KID indicating that $\mathbb{P}_r$ and $\mathbb{P}_g$ are closer to each other. In our evaluation, we use a polynomial kernel to calculate the KID.

Fig. 9 presents the results of our KID tests for different profiles of $b$. Notably, we observe that the cos-profile yields a lower KID, indicating better quality of generated samples compared to other profiles.

It is worth mentioning that another popular measure of similarity between two datasets of images, the Frechet Inception Distance (FID) Heusel et al. (2018), is often used to evaluate the quality of generated samples. However, FID is not applicable in our setting, as we intentionally work with a number of images smaller than the dimensionality of the input vector. In our experiments, the input vector has a dimension of 2048 (after passing through the Inception-v3), but we use only 1000 samples to estimate the covariance. Consequently, the covariance matrix becomes singular, making FID unsuitable for our evaluation. Therefore, we rely on KID as a robust alternative to assess the performance of our generative modeling approach.

### 3.7 Discussion

In this section, all the experiments presented and discussed were conducted under the "standard" setting of the Score-Based Diffusion (SBD). As a reminder, the standard setting involves training the score function on input data propagated with forward stochastic dynamics (1) from time 0 to $T$. Subsequently, synthetic data emerge through the propagation of an initially noisy (completely uninformative) image by the reverse stochastic process (2), which depends on the score function.

Our analysis of the standard setting reveals that generating high-quality synthetic data, and potentially enhancing their quality, does not necessitate initiating the inverse process at time $T$. This conclusion is supported by our examination of the auto-correlation function in the reverse process, as shown in Fig.(5), and the 1/2-correlation time, as illustrated in Fig.(6). Both figures indicate that the early stages of the reverse process do not contribute significantly to generating a synthetic image. Specifically, Fig.(6) demonstrates

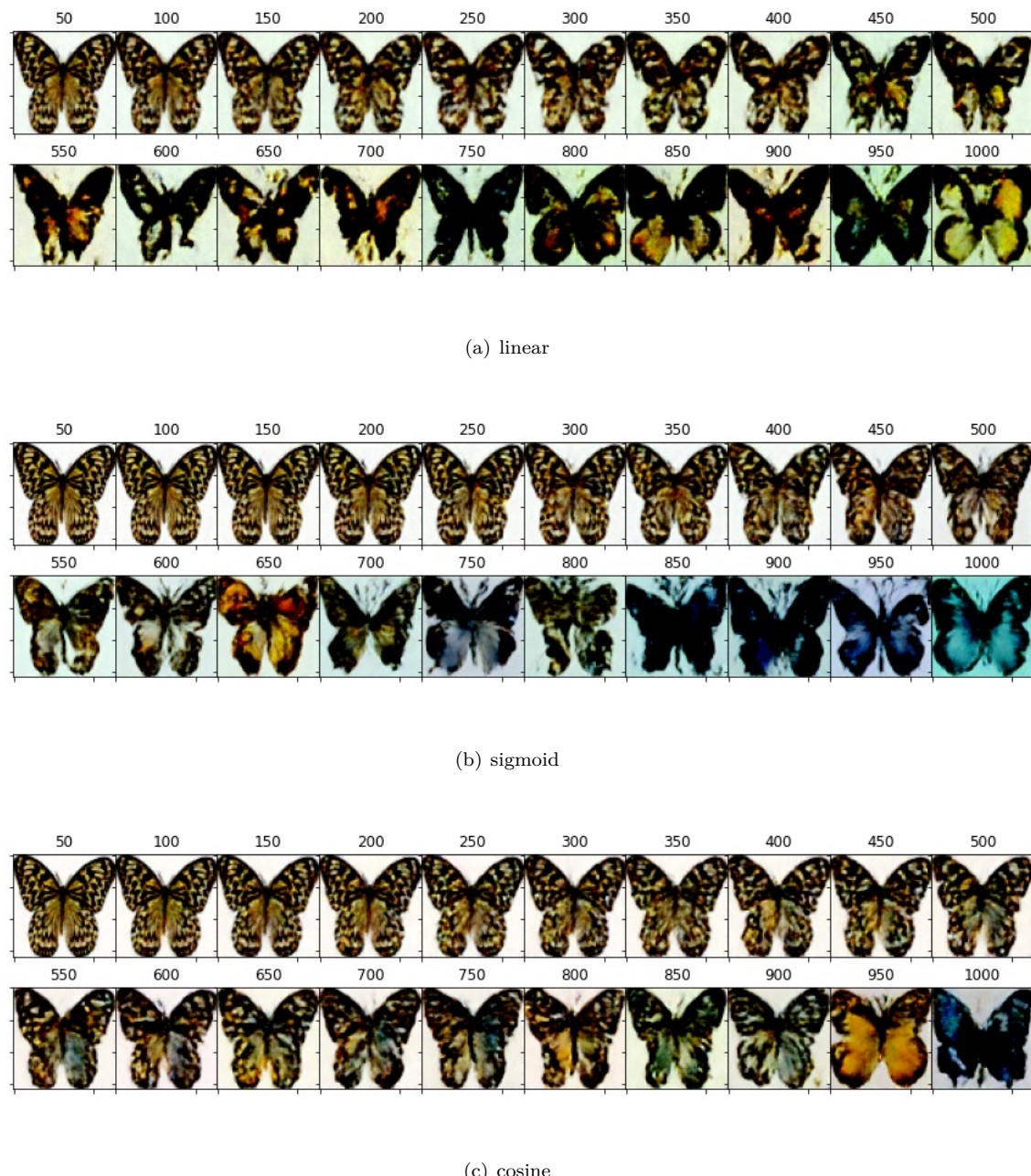

Figure 10: Synthetic images generated at $n = 0$ by making U-turn at different times from the $[0, 1000]$ range.

that correlations start forming not at $n = 1000$, but rather at $n \approx 200$ for linear $b$-profiles and at $n \approx 400$ for sigmoid- and cosine- $b$-profiles. Visual inspection of samples from the reverse dynamics, as displayed in Fig.(3), aligns with these findings. Similar observations and time scales are also evident in our reverse process KID score test, depicted in Fig. (9).

Considering the role of the score function itself as a function of time, which is extracted from the evolving data in the direct process, we inquire if it indicates when a synthetic image begins to form. Experimental evidence from Fig. (7b), depicting the evolution of the properly normalized norm of the score function,

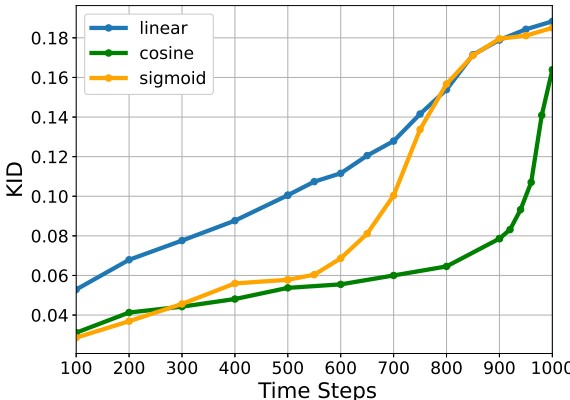

Figure 11: KID score plotted against $n$ for synthetic images generated with the U-turn conducted at the step $n$. Notably, we observe a distinct increase in the rate of KID versus $n$ growth at certain values of $n$, which we identify as the optimal positions for conducting the U-turn.

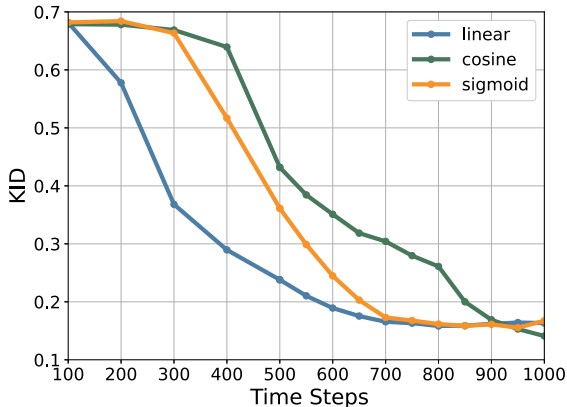

Figure 12: KID score for synthetic images generated (at $n = 0$) by initiation of the reverse process at the time $n$ with a random noise. We show KID as a function of the time $n$ of the reverse process random initiation.

indicates that the score function ceases to change at $n \approx 600$ for linear- and sigmoid- $b$-profiles, and at $n \approx 800$ for cosine- $b$-profiles.

Furthermore, the results of the KS test of Gaussianity in Fig. (8b) suggest that the reverse processes with linear-, sigmoid-, and cosine- $b$-profiles become largely Gaussian (proxy for uninformative) at $n \gtrsim 600$, $n \gtrsim 700$, and $n \gtrsim 900$, respectively. These findings collectively demonstrate that initiating the reverse process earlier than $n \approx 600$ for linear- and sigmoid- $b$-profiles, and $n \approx 800$ for cosine- $b$-profiles, does not significantly impact the quality of synthetic data.

Based on the findings discussed above, we have made significant advancements in our proposal for generating synthetic samples, which will be further elaborated in the following Section.

## 4  U-Turn Diffusion: Set Up and Numerical Experiments

We propose running a direct process, but with a shorter duration compared to the standard setting. Instead, we reverse the approach earlier, initiating the process in the opposite direction. We initialize the reverse

process using the last configuration from the forward process. This entire process, that is direct and reverse combined, is termed *U-Turn Diffusion*, emphasizing our expectation that direct process followed by U-Turn and the reverse process will ultimately produce a synthetic image. This synthetic image should, on one hand, closely resemble samples from the probability distribution representing the input data. On the other hand, it should be distinctly different from the original sample that initiated the preceding direct process when it arrives at $t = n = 0$.

---
**Algorithm 1** U-Turn

---
**Require:** $\nabla_{\mathbf{x}} \log p(\mathbf{x}, t)$, $\mathbf{x}_0 \sim p_{\text{data}}$
    1. Plot $S(t)$ using Eq. 18
    2. Find $T$ where $S(t)$ starts to flatten
    3. Calculate $\mathbf{x}_T$ using Eq. 7
    4. Initialize Eq. 2 at $t = T$ with $\mathbf{x}_T$
    5. Run the dynamaics until $t = 0$

---

The U-turn algorithm, as outlined in Algorithm 1, operates through a structured step-by-step process. It takes as input a dataset and a score function, which can be derived from either training or pre-trained models. To determine the ideal moment to execute the U-turn, we initiate the process by plotting the norm of the score function in the second step. This allows us to identify the point at which the score function begins to stabilize. Once this stabilization point is determined, we proceed to initialize the reverse dynamic at this moment using a noisy version of the actual images (as described in steps 3 and 4). Subsequently, we run the dynamics until it reaches the time $t = 0$, as outlined in step 5.

Synthetic images generated at $n = 0$ by making the U-turn at different times are shown in Fig. (10) for the three $b$-profiles. Consistently with the discussion above in Section 3.7 we observe, by examining the figures visually, that a principally new image of a high quality is generated if the U-turn occurs at $n \gtrsim 600$ for the linear-, sigmoid- $b$-protocols and at $n \gtrsim 850$ for the cosine- $b$-protocols.

The KID score, which compares synthetic images generated at $n = 0$ with the original data, is analyzed as a function of the U-turn time and presented in Fig. (11). The results displayed in the figures corroborate the observations made in Fig. (10). Notably, Fig. (11) reveals a significant finding – when the U-turn time surpasses an optimal threshold ($n \approx 600$ for linear and sigmoid $b$-profiles, and $n \approx 850$ for cosine $b$-profiles), the deterioration in the synthetic image quality accelerates considerably with increasing $n$ as compared to lower values. (Obviously, conducting a U-turn at a sufficiently small $n$ yields synthetic images at $n = 0$ that closely resemble the original images, resulting in a minimal KID.) In light of these observations, we deduce that these critical values of $n$, signifying a more rapid increase in KID with higher $n$, represent the optimal choices for the U-turn.

Fig. (12) showcases the outcomes of experiments analogous to those described earlier (leading to the results presented in Fig.(11)), however initiating the reverse process with a noise in this case. Evidently, in this case the reverse process does not retain any memory of the forward process, thus leading to increase in KID with decrease in $n$, where $n$ is the time step where we initialize the reverse process with the noise. Notably, the dependence of the KID on $n$ flattens as $n$ decreases. The flattening occurs around $n \approx 600$ for linear and sigmoid $b$-profiles, and around $n \approx 900$ for the cosine $b$-profile. This observation suggests that for random initialization of the reverse process, starting the process at $n = 1000$ is unnecessary. Instead, it is advantageous to start the reverse process at a smaller $n$, chosen based on the b-profile. Remarkably, a comparison between Fig.(11) and Fig.(12) underscores a notable advantage in initiating the reverse process with the final configuration from the forward process preceding the U-turn. This approach yields a marked reduction in KID, translating to an elevated quality of the synthetic image. For instance, examining Fig.(11), we find that the KID value for the sigmoid- $b$-profile process, with the U-turn executed at $n = 600$ (deemed the optimal U-turn point as discussed earlier), is approximately 0.07. In contrast, Fig.(12) demonstrates that random initiation of the reverse process at $n = 600$ leads to a significantly higher KID of about 0.19.

Two remarks are in order:

*Remark.* #1 Upon the recommendations of reviewers, we expanded our experiments from the Butterfly dataset (detailed in Appendix A) to encompass the CIFAR-10 dataset, with specifics outlined in Appendix B. The key takeaway from the CIFAR-10 evaluation is that our empirical findings, both regarding the basic and U-turn diffusion analyses, hold consistent with those observed in the Butterfly dataset.

*Remark.* #2 After completing this work, we discovered a related method called "boomerang," which was recently reported in Luzi et al. (2022). While there are some similarities between the boomerang and the U-turn diffusion described in this manuscript, it is essential to emphasize that they are distinct from each other. The boomerang method focuses on generating images that closely resemble the original samples, whereas the U-turn diffusion aims to create distinctly different images that approximate i.i.d. samples from the entire dataset. This fundamental difference also manifests in the applications suggested for the boomerang in Luzi et al. (2022), such as constructing privacy preserving datasets, data-augmentation and super-resolution. Given these distinctions, it would be intriguing to extend the analysis techniques we developed, including the auto-correlation functions, the score-function norm, KS criteria, and the KID metric, to the boomerang method and its interesting applications involving local sampling.

# 5 Related Papers

In this section, we follow the recommendation of an anonymous reviewer and provide a discussion of prior studies, emphasizing their relevance to the topics covered in this manuscript.

## 5.1 Jain & Poole (2022): Optimizing Sample Quality

Jain et al. (2022) proposed a clever strategy for improving sample quality in diffusion models. They showed that by focusing on estimating scores only in the early stages of the diffusion process ($t \in [0, 0.4]$) and employing a reduced number of discretization steps in the reverse stochastic differential equation (SDE), they could approach the desired quality achievable when considering all timesteps ($t \in [0, 1]$). To compensate for the error introduced by this approach, they ran Markov Chain Monte Carlo (MCMC) at each step, effectively pushing the samples towards the desired marginals. It's worth noting that this approach didn't significantly increase computational overhead, as the reduction in the number of steps was balanced by the need to run MCMC at each step.

## 5.2 Raya & Ambrogioni (2023): Early Stage Analysis

Raya et al. (2023) conducted a thorough analysis of diffusion models in the context of symmetry breaking. Their study revealed that the early stages of the reverse dynamics contributed minimally to generating new samples. Based on these observations, they proposed a "late start" strategy. While their idea aligns conceptually with the U-turn approach discussed in this manuscript, it's important to highlight that our justification for the timing of the late start is rooted in analysis insights rather than being solely empirical, as in their work.

## 5.3 Zheng et al. (2023); Franzese et al. (2023): Significance of Diffusion Time

The significance of diffusion time in basic score-based generative models was explored in Zheng et al. (2023) and Smith et al. (2023). These studies advocate using shorter diffusion times to enhance both training and sampling efficiency. Their approach involves employing auxiliary models to bridge the gap between the initial and final distributions in the diffusion process. Additionally, they introduce pre-trained generative models, such as Variational Autoencoders (VAEs) or Generative Adversarial Networks (GANs), to approximate non-Gaussian distributions in the reverse process. Moreover, Smith et al. (2023) posed the question of determining the optimal number of time steps as an optimization problem and argued that excessive diffusion time can have detrimental effects. This conclusion aligns with some of our ideas behind the U-turn algorithm.

### 5.4 Das et al. (2023): Noise Scheduling

Das et al. (2023) contributed to the topic of designing improved noise schedules. They initially studied the shortest path in the parameter space between two Gaussian distributions and subsequently applied these findings to fit the shortest path to data. Their claim is that this new schedule leads to better diffusion of data in both space and time when compared to conventional diffusion models.

### 5.5 De Bortoli et al. (2021): Schrödinger Bridge Approach

De Bortoli et al. (2021) introduced a Schrödinger Bridge (SB) based approach to facilitate the transformation between arbitrary probability measures within a finite time frame. They utilized the diffusion score matching approach to address the SB problem. While their resulting scheme requires fewer time steps compared to basic diffusion models, it necessitates multiple iterations between the two probability measures to converge.

### 5.6 Lipman et al. (2023): Generalized Markov Kernels

Lipman et al. (2023) proposed a generalization that removed the standard restriction on the Markov kernel, allowing for greater flexibility in selecting the path between two distributions. They chose to train the model using Optimal Transport conditional probabilities, resulting in more efficient paths compared to the popular VP Diffusion process and requiring fewer time steps.

In conclusion for this section, the ideas discussed in the aforementioned papers – such as ignoring the early stage of reverse dynamics, suggesting fewer steps, and other enhancements to computational efficiency – are likely to contribute to improvements in the initial stages of our U-turn approach. However, it is essential to emphasize that our contributions extend beyond the late start strategy; they also encompass the critical utilization of the last sample from the direct process before initiating the U-turn. This unique combination of techniques is critical to our overall conclusions regarding the advantages of the U-turn approach.

## 6 Conclusions and Path Forward

This paper delves into the analysis of popular Score-Based Diffusion Models (SBD), which are rooted in the idea that observed data are outcomes of dynamic processes. These models involve two related stochastic sub-processes: forward and reversed dynamics. The design of these models offers flexibility in choosing the advection and diffusion components of the dynamics. Three distinct advection-to-diffusion $b$-protocols, developed in prior publications on the subject, were adopted to explore this freedom. While the Fokker-Planck equations for the two sub-processes are equivalent, actual samples diverge as one advances forward and the other backward in time.

Our first accomplishment is extending analysis beyond single-time marginals to study time correlations through auto-correlation functions. This allowed quantification of information retention, by distributing in the score function, and than recovery of the information in the reverse process. The analysis unveiled diverse regimes, time scales, and their dependency on the chosen protocols.

The study then connects the decay patterns in auto-correlation functions to single-time objects, average of the weighted score-function 2-norm and the Kolmogorov-Smirnov metric. The temporal behaviors of these single-time objects are linked to the two-time correlation patterns, providing insights for potential control applications (see discussion below).

Informed by the temporal analysis of the SBD, a novel U-Turn Diffusion process, which is the climax of the manuscript, was devised, suggesting an optimal time to transition from forward to reverse sub-processes. The paper employs the KID test to assess the quality of U-Turn diffusion. Remarkably, the results demonstrate the existence of an optimal U-turn time for generating synthetic images which are of the best quality within the scheme.

In summary, this work thus not only advances our understanding of the SBD models but also offers a new U-Turn algorithm to enhance the quality of synthetically generated data.

The avenues for further exploration stemming from this study are delineated along three principal lines, each aiming to enhance further our understanding and application of the SBD models:

- *Fine-Tuning Protocols Using Time-Local Indicators*: Our immediate focus will be on optimizing and controlling the *b*-protocols to be data-adaptive. Employing time-local indicators such as the weighted average norm of the score-function and the KS test, we intend to align the *b*-protocols with the specific data characteristics.

- *Enhancing U-Turn enforced SBD with Data-Specific Dynamics:* Building on the success of the U-Turn enforced SBD approach, we aim to extend its utility by incorporating data-specific correlations and sparsity features into the underlying advection/diffusion dynamics. For instance, when initial data showcases spatial correlations, we plan to develop SBD techniques grounded in spatio-temporal stochastic partial differential equations.

- *Establishing Theoretical Connections to Non-Equilibrium Statistical Mechanics:* We intend to work on connecting the U-Turn enforced SBD approach to non-equilibrium statistical mechanics concepts, particularly those like the fluctuation theorem (e.g., Jarzynski and Crook relations) and Schrödinger bridge approaches. The exploration of this theoretical nexus, informed by existing literature and approaches Jarzynski (1997); Crooks (1999); Léonard & ,Modal-X. Université Paris Ouest, Bât. G, 200 av. de la République. 92001 Nanterre (2014); Chen et al. (2021); Sohl-Dickstein et al. (2015); De Bortoli et al. (2021), holds potential for illuminating the underlying mechanisms driving generative AI's power.

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

## A    Implementation Details

In our experimentation, we employ fundamental diffusion algorithms sourced from the Hugging Face Diffusers library von Platen et al. (2022). Our dataset consists of a collection of 1000 butterfly images, accessible via the Hugging Face Hub [1]. Since the images vary in dimensions, we uniformly resize them to $64 \times 64$.

To model the score-functions within the diffusion models, we adhere to the established approach of utilizing a U-net architecture. Detailed specifications of the U-net structure are available in the code (submitted along with the manuscript). In our computations, we adopt a batch size of 64, conduct 100 epochs, and leverage the Adam optimizer with a learning rate of $8 \times 10^{-5}$. For evaluating the KID, we make use of the implementation provided by TorchMetrics Detlefsen et al. (2022).

## B    CIFAR-10

Incorporated in light of feedback from anonymous reviewers, this Appendix seeks to validate the findings presented in the main manuscript using alternative and broader datasets, such as CIFAR-10. The goal is to assess the diversity of synthetic images, both quantitatively and qualitatively, on a data set which is more expressive (as buttterfly all look similar).

Subsequent sub-sections of the Appendix demonstrate that the newly introduce measure of diversity, as well as the behavior of the norm of the score function of the KID (our primary single-time indicators), are consistent with the findings reported in the main part of the paper.

Due to time constraints for addressing reviewer comments, we limit our discussion to training the diffusion model on CIFAR-10 for 2000 epochs (a relatively modest duration by current standards) and only discuss the linear $b$-protocol and the two indicators mentioned.

---

[1] `https://huggingface.co/datasets/huggan/smithsonian_butterflies_subset`

## B.1 Norm of the Score Function

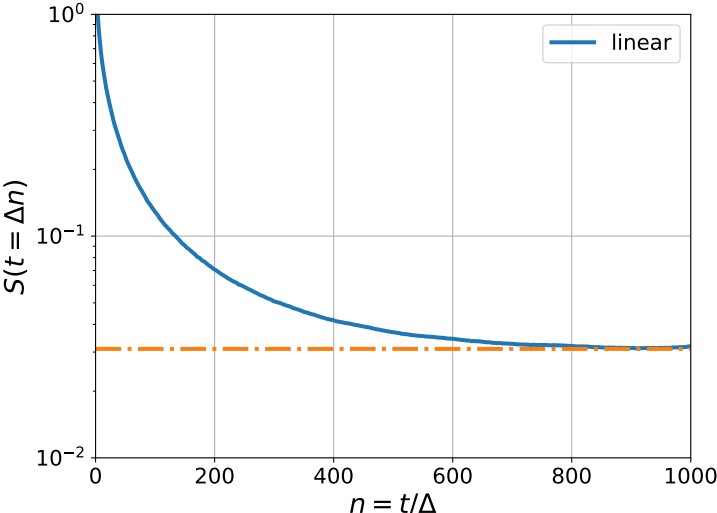

Figure 13: Norm of the score function for CIFAR-10.

Following Algorithm 1, the score function's norm is derived from Eq. 18. Figure 13 reveals a saturation point for the norm around $t_1 = 760$ – therefore suggesting that this is the optimal time for making the U-turn.

## B.2 Test of Diversity: Nearest Neighbor Test

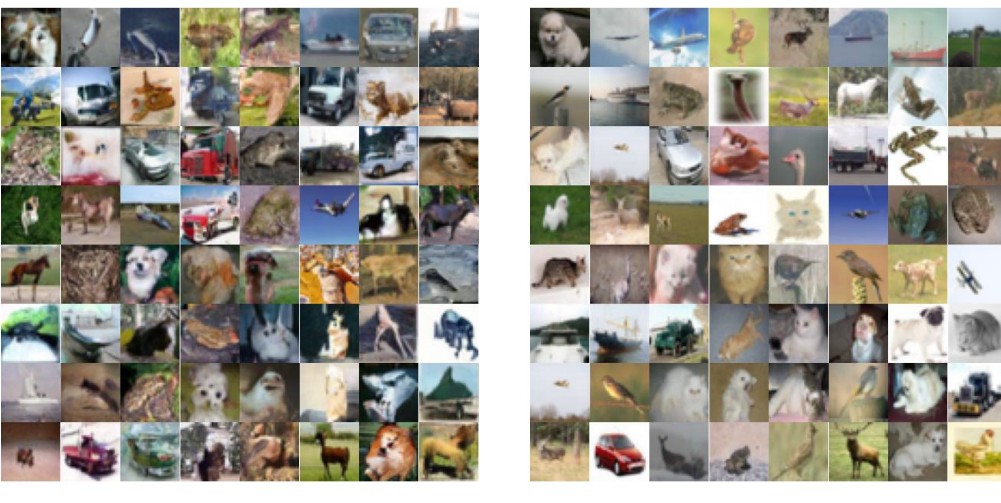

(a) Generated Images  (b) Nearest neighbors using cosine similarity 20.

Figure 14: U-Turn for $T = 500$

For a quantitative assessment, we applied the U-turn method for different $T$ values, examining the generated image diversity by comparing them to their nearest neighbors using the cosine similarity formula:

$$C_{\mathbf{x},\bar{\mathbf{x}}} = \frac{\langle \mathbf{x}, \bar{\mathbf{x}} \rangle}{\|\mathbf{x}\|_2 \|\bar{\mathbf{x}}\|_2}. \tag{20}$$

Here, $\mathbf{x}$ is an image from CIFAR-10, while $\bar{\mathbf{x}}$ is one generated by U-Turn.

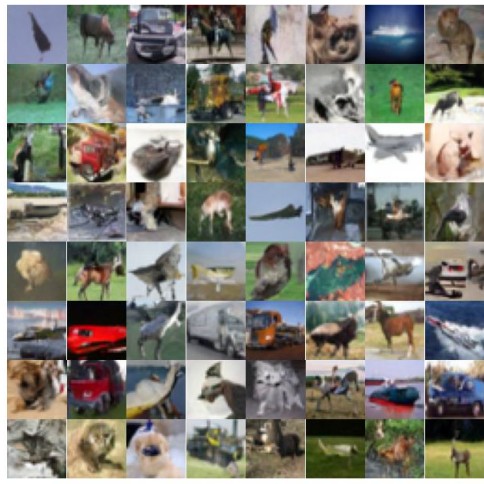 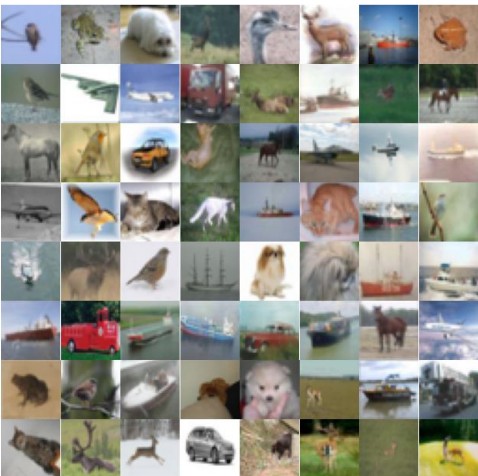

(a) generated                              (b) Nearest neighbors using cosine similarity 20.

Figure 15: U-Turn for $T = 600$

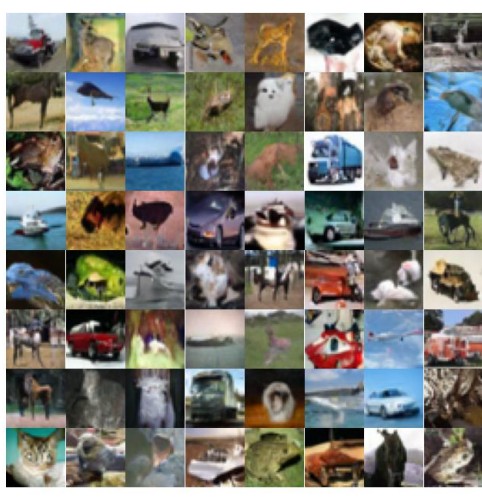 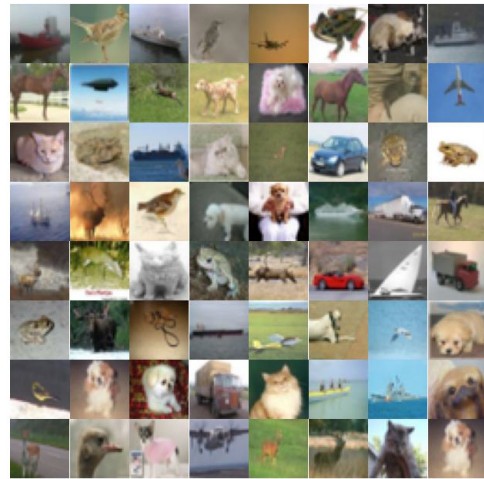

(a) Generated Images                       (b) Nearest neighbors using cosine similarity 20.

Figure 16: U-Turn for $T = 700$

Figures 14 through 19 illustrate our observations. Notably, applying U-turn before $t_1$ yields images with limited diversity (Figures 14 through 16). However, post-$t_1$ applications, as in Figure 17, result in a more varied set of generated images, as evident in Fig. 17 for $T = 800$. This increased diversity persists for subsequent $T$ values. As seen in Figs. 18, 19 the diversity of images generated at $T = 900$ and $T = 100$ is the same as $T = 800$, therefore reaffirming the U-turn method's capability in image generation.

## B.3   KID Test

Figure 20 showcases the KID scores across various time steps. These results echo our earlier observations with the butterfly dataset, where adding extra diffusion steps after reaching the optimal point deteriorates the KID. However, it's noteworthy that the KID scores remain relatively high due to the limited training epochs.

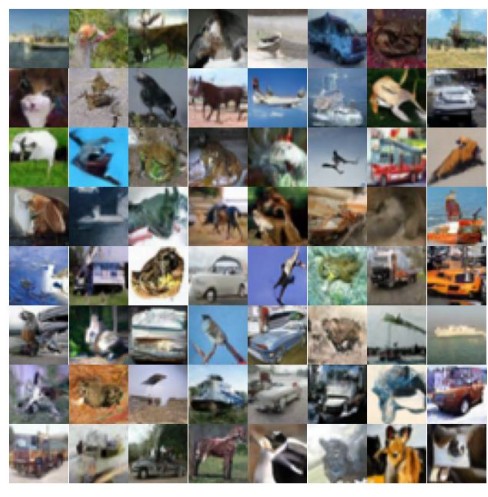 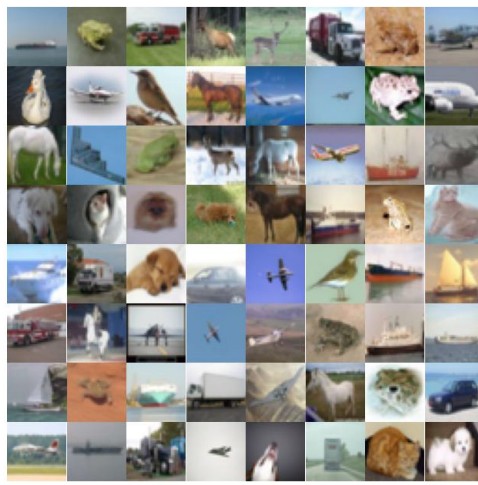

(a) Generated Images          (b) Nearest neighbors using cosine similarity 20.

Figure 17: U-Turn for $T = 800$

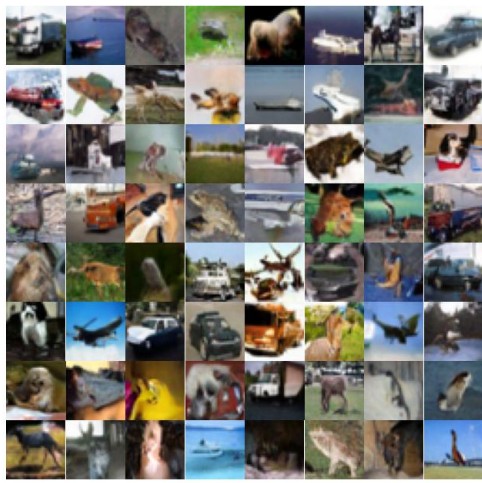 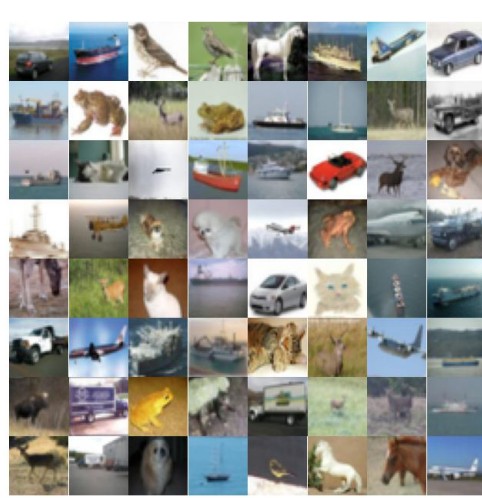

(a) Generated Images          (b) Nearest neighbors using cosine similarity 20.

Figure 18: U-Turn for $T = 900$

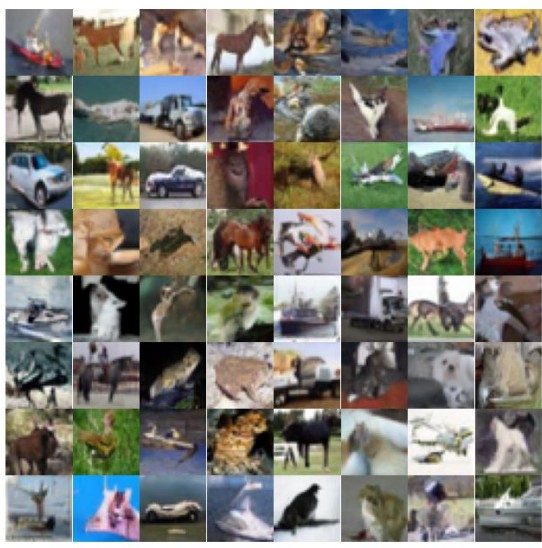 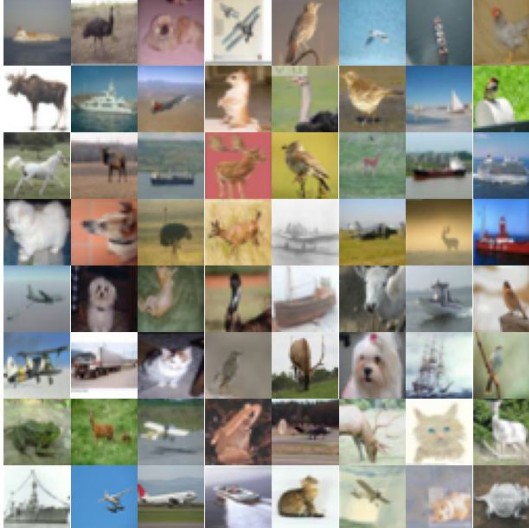

(a) Generated Images            (b) Nearest neighbors using cosine similarity 20.

Figure 19: U-Turn for $T = 1000$

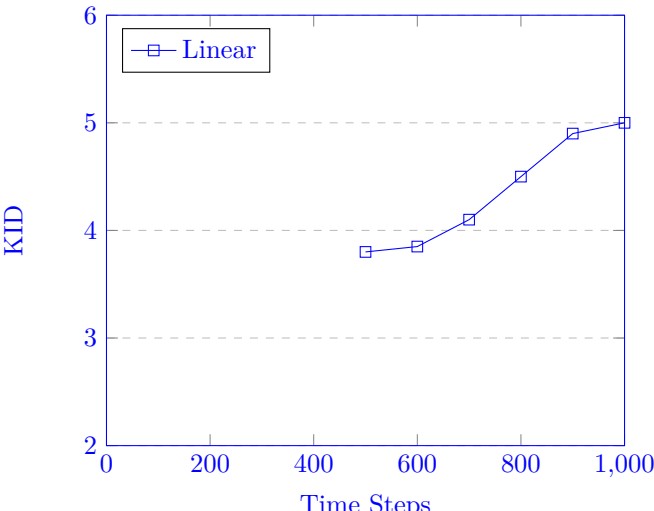

Figure 20: KID score for CIFAR-10 dataset. KID is computed for 50k generated images with the same number of real images from the dataset.

