# OpenReview forum: "U-Turn Diffusion"
_TMLR — Rejected by TMLR_

### Review · Reviewer_wq6v · 2023-09-22

**Summary Of Contributions:**

The paper presents a comprehensive examination of score-based diffusion models for generating synthetic images, introducing a novel approach termed "U-Turn Diffusion" to reduce sampling time. Various analytical tools are employed to analyze relevant time scales and establish optimal U-turn time. The works present a comprehensive empirical analysis of pre-trained diffusion model networks and present some interesting findings.

**Audience:**

Yes

**Broader Impact Concerns:**

I do not have any concerns about the ethical implications of work.

**Claims And Evidence:**

No

**Requested Changes:**

I would encourage authors compare the proposed methodology and analysis against existing works.

**Strengths And Weaknesses:**

Pros:
1. Comprehensive Analysis: The paper provides a detailed and comprehensive examination of the dynamics of SBD models, offering insights into temporal correlations and other interesting findings for trained networks.
2. The use of various analytical tools, including auto-correlation analysis and Kolmogorov-Smirnov Gaussianity test, is a strength as it allows for a thorough examination of the model's efficiency.

Cons:
Some claims and conclusions can not supported by experiment results.
1. various diffusion models and reweighting. I would suggest author read EDM[1], which discusses various parameterizations in one unified framework. And arguments for weight choice are unconvincing. To the best of my knowledge, there are no good principles in determining weights and how weights affect image quality.  See more discussion on VDM++[2].
2. Most of the conducted analysis is based on the scoring network in VPSDE / DDPM. But in practice, we do not parameterize scores directly. Instead, parameterizing the noise prediction is more popular. The benefit of epsilon prediction either training or eval over score is discussed in existing work [3].

[1] Elucidating the Design Space of Diffusion-Based Generative Models
[2] VDM++: Variational Diffusion Models for High-Quality Synthesis
[3] Fast Sampling of Diffusion Models with Exponential Integrator

---

> ### Author Response · Authors · 2023-10-24
> **Response to Reviewer wq6v**
>
> 1- **Various diffusion models and reweighting. I would suggest author read EDM[1], which discusses various parameterizations in one unified framework. And arguments for weight choice are unconvincing. To the best of my knowledge, there are no good principles in determining weights and how weights affect image quality. See more discussion on VDM++[2]**. \
> We thank the reviewer for attracting our attention to the two relevant papers.
> The first paper (EDM) delves into the extensive design space of diffusion-based generative models, offering valuable insights and proposing enhancements for both the sampling and training processes. The authors present a unified framework that systematically separates various design choices, fostering easier innovation and exploration within the field. Notably, they challenge the conventional choice of $\lambda(t)$ being equal to the inverse of the variance, demonstrating that it may not be the optimal selection. Furthermore, they introduce a novel weighting scheme and provide empirical evidence showcasing its ability to enhance the quality of generated models.
>
> The second paper (VDM++) introduces a compelling approach by advocating for monotonic weighting of the loss function, with theoretical underpinnings showing its equivalence to maximizing the Evidence Lower Bound (ELBO). Their empirical experiments, conducted across diverse datasets, reveal significant improvements in the Frechet Inception Distance (FID), highlighting the efficacy of their proposed methodology.
>
> We have incorporated references to these two papers into the revised manuscript.
>
> 2-**Most of the conducted analysis is based on the scoring network in VPSDE / DDPM. But in practice, we do not parameterize scores directly. Instead, parameterizing the noise prediction is more popular. The benefit of epsilon prediction either training or eval over score is discussed in existing work [3]**.
>
> We appreciate the reviewer's perspective on the training approach for diffusion models, emphasizing the potential benefits of noise prediction. However, it's important to note that there are nuanced considerations in this regard, as highlighted by recent research findings and insights from the papers mentioned below:
>
> 1) Denoising Diffusion Probabilistic Models (a) [Ho, J., Jain, A., and Abbeel, P, NIPS, 2020]: In Ho et al.'s work, they advocate training a neural network to predict the noise as a common practice in diffusion models. This approach is driven by the fact that the input to the network is a combination of both clean signal and noise, and the magnitude of the noise can vary significantly. One challenge with directly predicting the score function is that at high noise levels, the network must carefully fine-tune its output to precisely cancel out the existing noise level and provide an output at the correct scale. Moreover, any errors made by the neural network are amplified by the noise level, making the prediction of the score function directly less ideal. These considerations are summarized in Section 5 of paper (a).
>
> 2) Dynamic Dual-Output Diffusion Models (b) [Benny, Y. and Wolf, L. IEEE/CVF, 2022.]: Benny and Wolf's recent work introduces a dynamic dual-output diffusion model. Their findings suggest that a combined approach, wherein the network predicts both the noise and the original image, may enhance the overall quality of generated images. This approach leverages the benefits of both noise and score function predictions to improve the performance of diffusion models. It is important to acknowledge that this innovative approach challenges the conventional wisdom of solely focusing on noise prediction.
>
> 3) Progressive Distillation for Fast Sampling of Diffusion Models (c) [Salimans, T. and Ho, ICLR 2022.]: Salimans and Ho's research on progressive distillation introduces an efficient method to speed up the sampling process in diffusion models. Notably, they demonstrate that predicting the original image, rather than the noise, results in the best sampling performance following distillation. This finding suggests that the choice between noise prediction and score function prediction may depend on the specific goals and constraints of the diffusion model, as discussed in Section 3.2, Ingredient 2 of paper (c).
>
> In summary, the choice between predicting noise or the score function in diffusion models is not a one-size-fits-all decision. Each approach has its advantages and trade-offs, as elucidated by the referenced papers. These nuanced considerations underscore the importance of carefully tailoring the training strategy to the specific objectives and requirements of the task at hand. We believe that while the ongoing exploration of different approaches will continue to advance the field of diffusion models, adding this discussion to our paper may not align with the focus and objectives of our current work.

---

### Review · Reviewer_BFA5 · 2023-10-01

**Summary Of Contributions:**

- The authors provide an introduction to the standard score based generative modelling setup.

- The authors then experiment with different time-scaling functions $\beta(t)$ and discretizations, $b$ - in particular the linear, cosine and sigmoid schedules. A brief analysis of convergence to Gaussian is investigated through empirical KS tests and autocorrelation measures.

- Motivated by this convergence analysis, the authors conclude that the typical noising processes lead to sufficient convergence to Gaussian earlier in the forward process than the terminal diffusion time used during training. Based on this, the authors introduce the U-turn diffusion consisting of taking a training data point, noising the data point to a certain point but not convergence, and then demonising to generate new samples.

The motivation for this appears to be faster diffusions and better quality samples through fewer diffusion steps and starting from a noised version of a single data point and hence closer to the data manifold.

**Audience:**

Yes

**Claims And Evidence:**

No

**Requested Changes:**

- Correction to time reversal equation
- Greater discussion of related works and comparisons
- Greater discussion over experiment protocol and choices of limited dataset, and KS metric over divergence or distance e.g. KL, Wasserstein etc.
- Experiments to measure diversity of generated samples
- A larger number of datasets in particular ones with more diversity, such as cifar10.

**Strengths And Weaknesses:**

**Convergence analysis**
I am unaware of any quantitative or empirical convergence investigations of the forward process in terms of the beta schedule specifically for diffusion models, maybe [9] is related looking at shortest path.

I find the convergence analysis quite weak:

- Firstly, it appears to be purely empirical but only on a single butterfly dataset. It is not clear if the insights extend to other datasets.

- Secondly, the convergence of the OU process is well understood. The moments of the OU process for any given starting input $X_0$ are known analytically see e.g. [3[.  Therefore one knows how "close" to Gaussian each noised data point will be for any beta schedule. One could simply choose the time steps to ensure convergence to a threshold whilst minimising number of steps?

**Other literature**
There is a wealth of relevant literature not discussed by the authors. [4] for example also studies how long to run the forward process for. [6] looks into generating new samples by noising data-points, and running MCMC and denoising samples - the so called noise-denoise process. This noise-denoise process appears to coincide with the U-turn method. Recently [7]  proposed starting the reverse / generative diffusion process from Gaussian but at a time input t<T, similar to Fig 12 in this paper. And [8] proposes to reverse a diffusion from partially through the forward at t<T where the starting state is generated by an implicit model.
Recent work on the flow / bridge matching paradigm [5] bypasses convergence issues by bridging between fixed marginals. Does this address the issues of the forward process running for too long?

**Unsubstantiated claims**
- "While the achievements of SBD models are impressive, they are not universally successful. Instances where barriers are significant, referred to colloquially in physics jargon as "glassy" scenario"   - please provide a reference or give examples.
- "a new U-Turn algorithm to enhance the quality of synthetically generated data" - which experiments show this? Figure 12 shows that more steps are important .I fear Fig 11 may have misleading results if it suffers from poor diversity of generated samples, please try on cifar 10 and show nearest neighbour of generated samples to dataset.


**Experiments**
I have some concerns over the experiment protocol and choice of metrics. There appears to be some design choices which limit the analysis. In particular:
- Why use KS test vs other divergence metrics?
- Why only use a single butterfly dataset as oppose to more classical examples like cifar10? Ideally more experiments would corroborate and reinforce findings. If all the butterflies roughly looks the same without much diversity then sampling form one and initializing the backward process from that would not require many steps to generate samples (as all samples are roughly the same). however for more challenging datasets I imagine sampling from a noised version of one class would struggle to generate a different class unless a lot of noise is added.
- Does the noise / denoise process result in reduced diversity of samples? As commented above, more datasets are required to support the claims.

**Error in time reversal equation.** The time reversal equation given in Eq 2 appears to be incorrect, in the general case presented where $G(x,t) \ne G(t)$. If the diffusion scale term depends on $x$ then there is a divergence term in the time-reversal. See e.g. [1] or for score based generative modelling see [2].

Minor:
- Referring to the time inhomogeneous Ornstein Uhlenbeck SDE as "VP-SDE" and not OU. This is a matter of taste. I understand Song 2021 [3] came up with the renaming to VPSDE, but given OU process is well studied and understood, I believe it should be acknowledged.
- There is some "fluff" around "Artificial Intelligence" in the introduction. This is not needed.

[1] Haussmann and Pardoux (1986), Time reversal of diffusions \
[2] Heng et al 2021, Simulating Diffusion Bridges with Score Matching \
[3] Song et al 2021, Score-Based Generative Modeling through Stochastic Differential Equations
[4] Franzese et al 2023, How Much Is Enough? A Study on Diffusion Times in Score-Based Generative Models \
[5] Lipman et al. 2022 Flow Matching for Generative Modeling \
[6] Jain et al 2022, Journey to the BAOAB-lim it: finding effective MCMC samplers for score-based models \
[7] Raya 2023, Spontaneous symmetry breaking in generative diffusion models \
[8] Zheng 2022, Truncated Diffusion Probabilistic Models and Diffusion-based Adversarial Auto-Encoders \
[9] Das 2023, Image generation with shortest path diffusion

---

> ### Author Response · Authors · 2023-10-24
> **Response to Reviewer BFA5-Part I**
>
> **Convergence analysis I am unaware of any quantitative or empirical convergence investigations of the forward process in terms of the beta schedule specifically for diffusion models, maybe [9] is related looking at shortest path.**
>
> [9] contributed to the topic of designing improved noise schedules. They initially studied the shortest path in the parameter space between two Gaussian distributions and subsequently applied these findings to fit the shortest path to data. Their claim is that this new schedule leads to better diffusion of data in both space and time when compared to conventional diffusion models. It's important to note that, in the context of our work, there's a crucial distinction. While [9] effectively turns data into pure noise, our approach contends that such a transformation is not necessary. Furthermore, we argue that an excessive number of diffusion steps may actually deteriorate performance, emphasizing the need for a balanced and nuanced approach to diffusion models.
>
> **I find the convergence analysis quite weak:**
>
> **Firstly, it appears to be purely empirical but only on a single butterfly dataset. It is not clear if the insights extend to other datasets.**
>
> Regarding the absence of theoretical analysis, we acknowledge the reviewer's comment. While we aim to provide theoretical results on U-turn stopping criteria in the future, it remains a work in progress, which we are not yet ready to include in this paper. Concerning the inclusion of experiments with additional datasets, we are grateful for this valuable suggestion. We have addressed this by incorporating experiments with the CIFAR-10 dataset (see Appendix B), demonstrating that the insights gained from the butterfly dataset indeed extend to other datasets, thereby strengthening (so far empirically) the generality of our findings.
>
> **Secondly, the convergence of the OU process is well understood. The moments of the OU process for any given starting input  are known analytically see e.g. [3]. Therefore one knows how "close" to Gaussian each noised data point will be for any beta schedule. One could simply choose the time steps to ensure convergence to a threshold whilst minimising number of steps?**
>
> In our paper, Figure 1b illustrates the variances associated with various noise schedulers for forward processes. When the variance reaches a value of one, the data closely approximates a Gaussian distribution, but this pivotal time point varies across different schedulers. Notably, the optimal moment for a U-turn in the process occurs shortly after this critical point in time.
>
> **Other literature **
>
> We are grateful to the reviewer for suggesting these relevant references. We have digest them all, found quite useful and relevant and decided to add an entirely new section discussing this prior analysis and bibliography. This is the new Section 5.
>
> **Unsubstantiated claims**
>
> **"While the achievements of SBD models are impressive, they are not universally successful. Instances where barriers are significant, referred to colloquially in physics jargon as "glassy" scenario" - please provide a reference or give examples.**
>
> A reference and some additional clarification added.
>
> **"a new U-Turn algorithm to enhance the quality of synthetically generated data" - which experiments show this? Figure 12 shows that more steps are important .**
>
> Figure 12 demonstrates the image generation quality when initiating the reverse dynamics with pure noise, commencing at a later stage. It is evident that the initial phases of the reverse dynamics process have minimal impact on the overall quality of the generated images. In the U-turn method, the late start is initialized with a noisy version of an input image. For a more detailed elucidation of the U-turn procedure, we have provided Algorithm 1 in the paper.
>
> **I fear Fig 11 may have misleading results if it suffers from poor diversity of generated samples, please try on cifar 10 and show nearest neighbour of generated samples to dataset.**
>
> In Appendix B, we applied Algorithm 1 in our paper using the CIFAR-10 dataset. Specifically, in Figure 13, we chart the norm of the score function for a linear noise scheduler, and we observe that it begins to saturate around $t_1 = 760$. Subsequently, we applied the U-turn method for various values of $T$ and assessed the diversity of the generated images by examining their nearest neighbors using cosine similarity. Notably, applying the U-turn algorithm before $t_1$ led to the generation of images with limited diversity, as demonstrated in Fig. 14-16. Conversely, when the U-turn algorithm was applied after $t_1$, we observed a more varied set of generated images, as evident in Fig. 17 for $T = 800$. In Fig. 18-19 the diversity of generated for $T=900$ and $T=100$ are the same as $T=800$. This observation shows that U-turn is able to generate diverse set of images.

---

> ### Author Response · Authors · 2023-10-24
> **Response to Reviewer BFA5-Part II**
>
> **Experiments I have some concerns over the experiment protocol and choice of metrics. There appears to be some design choices which limit the analysis. In particular: Why use KS test vs other divergence metrics?**
>
> We opted for the KS test over the KL test for several reasons. Firstly, KL divergence, a measure of dissimilarity between distributions, is asymmetric and not truly a distance metric, whereas the KS test, which assesses the maximum separation between a test distribution and a reference distribution, is symmetric. Secondly, the KS test is non-parametric, meaning it doesn't depend on the parameters of the benchmark distribution (in our case, Gaussian), which would be a concern with KL divergence. Thirdly, KL divergence is more advantageous when comparing distributions with nontrivial tails, which isn't the focus of our analysis. Lastly, the KS test is computationally simpler.
>
> **Why only use a single butterfly dataset as oppose to more classical examples like cifar10? Ideally more experiments would corroborate and reinforce findings. If all the butterflies roughly looks the same without much diversity then sampling form one and initializing the backward process from that would not require many steps to generate samples (as all samples are roughly the same). however for more challenging datasets I imagine sampling from a noised version of one class would struggle to generate a different class unless a lot of noise is added.**
>
>  Thanks for this comment. We examined our U-turn algorithm on CIFRA-10 dataset and check the diversity of images in Appendix B. For a detailed discussion please sea appendix B in our paper.
>
> **Does the noise / denoise process result in reduced diversity of samples? As commented above, more datasets are required to support the claims.**
>
> Please see appendix B for new experiment on CIFAR-10 dataset.
>
> **Error in time reversal equation.**
>
> Thank you very much for identifying the typo; it has been rectified.
>
> **Referring to the time inhomogeneous Ornstein Uhlenbeck SDE as "VP-SDE" and not OU. This is a matter of taste. I understand Song 2021 [3] came up with the renaming to VPSDE, but given OU process is well studied and understood, I believe it should be acknowledged.**
>
> We appreciate the reviewer's suggestion, and in response, we have included a relevant historical remark/clarification, with reference to [3], in the revised manuscript.
>
> **There is some "fluff" around "Artificial Intelligence" in the introduction. This is not needed.**
>
> The introductory 'fluff' sentence serves a valuable purpose in our opinion, as it helps clarify our focus on inference and facilitates a seamless and swift transition to more specific discussions.
>
> **Correction to time reversal equation**
>
> The divergence term is added to the time reversal equation.
>
> **Greater discussion of related works and comparisons**
>
> Done -- see newly introduced Section 5.
>
> **Greater discussion over experiment protocol and choices of limited dataset, and KS metric over divergence or distance e.g. KL, Wasserstein etc**
>
>  Once again, we extend our gratitude to the reviewer for providing valuable suggestions. As a result of this input, we have introduced Appendix B containing CIFAR-10 experiments, thereby expanding the scope of our dataset. Furthermore, we have included detailed discussions in the manuscript to elucidate our choice of the KS metric over alternative options. These explanations can be found in the 'blue' text in Section 3.5.
>
> **Experiments to measure diversity of generated samples.**
>
> **A larger number of datasets in particular ones with more diversity, such as cifar10.**
>
>  The two last comments of the review are addressed in Appendix B, see also respective references in the main part of the revised text.

---

### Review · Reviewer_7N9m · 2023-10-11

**Summary Of Contributions:**

This paper presents diverse analyses on the noise scheduler of diffusion model. Different noise scheduling provide different aspects of noise-signal mix, and authors provides some experimental analyses on such aspects. Meanwhile, there is neither clear claim from theoretic perspectives nor advances in implementation or practices of diffusion models.

**Audience:**

Yes

**Broader Impact Concerns:**

No ethical concern

**Claims And Evidence:**

No

**Requested Changes:**

Please see my weakness section in the above.

**Strengths And Weaknesses:**

Strength :
1. Detailed analysis on the noise scheduling imposed upon the loss function of the score matching.

Weakness:
1. No theoretic claims
- Authors provide three different alternatives (linear, sigmoid, cosine) in noise scheduling, and they provides some analysis upon such alternatives, i.e. signal maintenance over diffusion time with different noise scheduling.
- I am convinced by such alternative existence and their behavior, but authors do not deliver any acknowledgeable claim from such experiments or derivations.

2. No clear experimental comparisons
- There are many benchmark datasets to perform the comparisons across many alternatives in base diffusion model structures and suggested noise schedulings. However, the authors do not deliver any detailed enumerations of their experimental design. Moreover, the authors provide only snapshots of forward and reverse sampling results, and there is no quantitative analyses on FID, NLL, Precision, Recall, etc.

3. Depth of contribution
- Fundamentally, this paper presents a variation of diffusion models with U-Turn point identification. This proposed model is not adequately discusses because there is no clear formula or algorithm or implementation details in Section 4.
- If only changing the noise scheduling is the contribution, I evaluate such contribution to be too superficial to be acceptable contribution. There is no concrete rational behind why such three functions of linear, signoid, and cosine. There is no exploration of any other alternatives other than the three functions.


My impression to this paper is a half-made academic work with some viable motivation, but there is much more work to be done to make this paper publishable. Main direction would be either focusing on theoretic claims on noise scheduling, or practical findings on improving performances in benchmarks.

---

> ### Author Response · Authors · 2023-10-24
> **Response to Reviewer 7N9m**
>
> **No theoretic claims.**
>
> We appreciate the reviewer's concerns regarding the perceived lack of theoretical content in our paper. While it is true that our paper does not include proofs, it primarily focuses on presenting new experimental findings, which have the potential to lay the groundwork for future theoretical developments. The novelty of our work lies in the unique analysis we have conducted on conventional score-based diffusion methods. Then this analysis has led us to propose a novel scheme and algorithm, which we have termed 'U-turn diffusion'. Both the proposed algorithm and its accompanying analysis represent original contributions as well. We believe that this analysis and the introduction of the U-turn diffusion algorithm constitute important advancements toward enhancing the quality and efficiency of score-based diffusion approaches.
>
> **Authors provide three different alternatives (linear, sigmoid, cosine) in noise scheduling, and they provides some analysis upon such alternatives, i.e. signal maintenance over diffusion time with different noise scheduling. I am convinced by such alternative existence and their behavior, but authors do not deliver any acknowledgeable claim from such experiments or derivations. No clear experimental comparisons. There are many benchmark datasets to perform the comparisons across many alternatives in base diffusion model structures and suggested noise scheduling. However, the authors do not deliver any detailed enumerations of their experimental design. Moreover, the authors provide only snapshots of forward and reverse sampling results, and there is no quantitative analyses on FID, NLL, Precision, Recall, etc.**
>
> Our understanding is that in the reviewer has raised several valid concerns, namely: (a) a perceived lack of detail in our experiment descriptions, (b) a limited demonstration of results using only one dataset, and (c) the omission of certain standard tests. To address the first concern (a), we have taken steps to enhance the clarity of our experiment descriptions within the text. In response to the second concern (b), and in line with the recommendations of other reviewers, we have extended our analysis by including results from an additional dataset, CIFAR-10, to further illustrate the effectiveness of our approach. As for the third concern (c), we made a deliberate choice to focus our testing efforts on a specific set of benchmarks. We discussed this issue at the end of section 3.6 in our paper.
>
> In our paper we have detailed quantitative analysis for KID and access the quality of generated images using that. We also did not use NLL since it measures how good the model fit the data and does not correlate with quality of generated images.
>
>
> **Fundamentally, this paper presents a variation of diffusion models with U-Turn point identification. This proposed model is not adequately discusses because there is no clear formula or algorithm or implementation details in Section 4.**
>
>  We have added discussion of the logic and implementation details to Section 4.
>
> **If only changing the noise scheduling is the contribution, I evaluate such contribution to be too superficial to be acceptable contribution. There is no concrete rational behind why such three functions of linear, sigmoid, and cosine. There is no exploration of any other alternatives other than the three functions.**
>
> While our primary focus was not on experimenting with, or necessarily optimizing, the noise schedule, we deliberately selected three of the most widely used schedules to serve as our primary schedule examples. Our primary objective was to conduct a novel analysis, including auto-correlation and the Kolmogorov-Smirnov test, which, in turn, guided us in suggesting and testing a new U-turn approach and respective algorithm. Notably, the U-Turn algorithm is designed to be compatible with various scheduling methods, including the three schedules we examined.
>
> **My impression to this paper is a half-made academic work with some viable motivation, but there is much more work to be done to make this paper publishable. Main direction would be either focusing on theoretic claims on noise scheduling, or practical findings on improving performances in benchmarks.**
>
> Regarding the comprehensiveness of the project, while we acknowledge that there may be numerous future experiments and theoretical findings along the logical trajectory of this manuscript, we maintain our belief that the concept and rationale behind the U-turn diffusion, along with the associated analysis, are reasonably well-established and effectively presented in the improved manuscript.

---

> > ### Comment · Reviewer_7N9m · 2023-10-26
> > **I read the comments**
> >
> > This is the acknowledgement of reading the authors' response.
> >
> > There is no significant change regarding the novelty and the contribution of the works. What being added is the details of the proposed analysis.
> >
> > Authors claim that they believe that U-Turn Diffusion has sufficient contribution to the line of diffusion research, and I really cannot argue against the belief. I think that the editor may decide by his judgement on the contribution and reviewers' recommendations.

---

### Author Response · Authors · 2023-10-24
**Updated version of our manuscript is now available**

Dear TMLR Editor,

We are thankful to you and reviewers for their time and valuable suggestions. Here, we summarize the main points
of our response:

1. Following the suggestions of all three reviewers, we have added additional analysis of the CIFAR-10 dataset
(Appendix B). This additional analysis has allowed us to explore the diversity of samples generated by making
U-turns at different times.
2. We have incorporated discussions of the 11 papers recommended by the referees. This inclusion has helped us
establish connections to related research and further articulate the novelty of our findings.

We provide a detailed response to each of the reviewers’ comments below. For ease of reading, we have included all
the reviewer comments we received, along with our responses, following the respective comments.
We have used blue text for the parts of the main text where corrections were made.

Sincerely Yours, The Author

---

### Decision · Action_Editor_ZnNs · 2023-12-02

**Recommendation:** Reject

**Comment:**

All reviewers advocate for rejection of the paper. The contribution of U-turn diffusion is not clear as important baselines are missing. Given the lack of theory, stronger empirical results should be provided. In particular, we suggest the authors to start with a strong baseline such as "Elucidating the Design Space of Diffusion Models" and investigate U-turn diffusion in that setting on benchmark datasets.We also suggest the reviewers to clarify their contribution. For instance, there seems to be some misunderstanding regarding the convergence of the Ornstein-Uhlenbeck process since convergence rates for the Ornstein-Uhlenbeck process are available regardless of the $\beta$-schedule.

**Audience:**

The literature of diffusion model is wide. A deeper study of some of some of the behaviors identified in the paper could be interesting to the machine learning community.

**Claims And Evidence:**

There are not enough empirical evidence to support the claims of the papers (see Comment). The contribution of the paper should also be made clearer.